# AdaS: Adaptive Gradient Descent for Spiking Transformers

Zijian Zhou [1]   Honglin Cao [1]   Ammar Belatreche [2]   Wenjie Wei [1]   Yimeng Shan [1]   Yu Liang [1]   Yu Yang [1]   Shuai Wang [1]   Yalan Ye [1]   Malu Zhang [1 3]   Yang Yang [1]   Haizhou Li [3 4]

## Abstract

Transformer-based Spiking Neural Networks (SNNs) combine Transformer performance with SNN energy efficiency through an event-driven self-attention mechanism. However, Spiking Transformers still lag behind their Artificial Neural Network (ANN) counterparts. Most existing studies address this issue through new architectural designs, yet few have explored optimization algorithms tailored to Spiking Transformers. We substantiate the excessive noise problem in Spiking Transformer training by quantitatively defining parameter-update noise and, based on this definition, providing theoretical analysis and experimental validation. To address this problem, we propose AdaS, an adaptive gradient descent method for Spiking Transformers. AdaS reduces excessive noise by adaptively incorporating a gradient update component into adaptive optimization. Instead of simply removing noise, AdaS maintains it at an appropriate level to preserve its generalization benefits, thereby improving the performance of Spiking Transformers. We conduct extensive experiments on various Spiking Transformer architectures and datasets from both computer vision and natural language processing. The results demonstrate that the proposed AdaS consistently enhances performance across different Spiking Transformers, validating its effectiveness and generalizability. This work is among the first systematic studies of optimizer design specifically for Spiking Transformers, offering a practical tool to narrow the accuracy gap with ANNs while preserving the energy advantages of spike-based computation. Code is available at https://github.com/CayleyZ/AdaS.

## 1. Introduction

Spiking Neural Networks (SNNs) have garnered more attention as third-generation artificial neural networks, owing to their energy efficiency and biological plausibility (Maass, 1997). These networks employ binary spikes as the fundamental units for information transmission, operating in a sparse spike-driven paradigm (Zhang et al., 2021b; Tang et al., 2024). The sparse synaptic communication characteristic of SNNs transforms conventional multiply-accumulate (MAC) operations into simpler accumulate (AC) operations, substantially enhancing computational efficiency (Li et al., 2024; Xu et al., 2024; Wei et al., 2025; Liu et al., 2026). This energy-efficient property has catalyzed the development of specialized neuromorphic hardware implementations, including SpiNNaker (Painkras et al., 2013), TrueNorth (Akopyan et al., 2015), Loihi (Davies et al., 2018), and Tianjic (Pei et al., 2019). However, existing spiking models typically suffer from certain performance limitations, which continue to constrain their widespread application.

The Transformer (Vaswani et al., 2017; Dosovitskiy et al., 2021) has shown exceptional performance across diverse domains through its self-attention mechanism for capturing long-range dependencies, establishing itself as the foundational architecture for large-scale pre-trained models. The integration of SNNs with Transformer architectures achieves an optimal balance between energy efficiency and computational performance, combining the low power advantages of spiking neural networks with the superior feature extraction capabilities of Transformers (Zhou et al., 2024; Zhang et al., 2025b). However, Spiking Transformers still underperform compared to their ANN counterparts, partially because existing approaches only focus on architectural innovations while directly adopting conventional training methods, overlooking the unique challenges of training SNNs (Wang et al., 2025b; Yao et al., 2024; Wei et al., 2026).

Currently, Transformer models are typically trained using adaptive optimization algorithms (e.g., AdamW) (Zhou et al., 2023b; Cao et al., 2025). Compared to traditional stochastic gradient descent (SGD), these adaptive methods introduce a certain amount of noise during parameter updates, which helps the model escape sharp local extrema and move into flatter regions of the loss function, thereby

---

[1]University of Electronic Science and Technology of China [2]Northumbria University [3]Shenzhen Loop Area Institute [4]The Chinese University of Hong Kong, Shenzhen (CUHK-Shenzhen). Correspondence to: Malu Zhang <maluzhang@uestc.edu.cn>.

*Proceedings of the 43rd International Conference on Machine Learning*, Seoul, South Korea. PMLR 306, 2026. Copyright 2026 by the author(s).

enhancing generalization (Chen et al., 2023). However, it is noteworthy that due to the non-differentiability of spiking neurons, Spiking Transformers cannot be trained directly and must rely on surrogate gradient functions to approximate the true gradient of the spike step function. This approximation introduces extra noise into the parameter update process (Qin et al., 2020). When the noise from the surrogate gradients is compounded with that inherent in the adaptive optimizer, the actual parameter update deviates significantly from the negative-gradient update. Such deviation can result in slower convergence, training oscillations, and reduced stability, ultimately affecting the model's performance (Sutskever et al., 2013). We refer to this as the excessive noise problem.

To substantiate the excessive noise problem in Spiking Transformer training, we first introduce a quantitative definition of parameter-update noise and then provide theoretical analysis and experimental validation based on this definition. To address the issue, we design an optimization algorithm named AdaS, which aims to adjust the overall parameter-update noise to an appropriate level by diminishing the noise introduced by the adaptive optimizer. Specifically, AdaS incorporates a momentum-based gradient update component into the adaptive update computed by the base optimizer and balances the contributions of the two update components through an adaptive weighting mechanism. Furthermore, the proposed AdaS algorithm offers two significant advantages. First, by utilizing the first-order momentum stored in the adaptive optimizer as an approximate estimate of the gradient-based update component, no additional memory overhead is incurred. Second, it can be seamlessly integrated into most adaptive optimizers, demonstrating high versatility and flexibility as a meta optimization algorithm. The main contributions of this paper are as follows:

- We identify the excessive noise problem in spiking transformer training induced by surrogate gradient learning. To substantiate it, we introduce a quantitative definition of parameter-update noise and provide analysis with experimental validation.

- We propose AdaS, an optimizer designed for SNNs. AdaS adjusts training noise to an appropriate level by combining adaptive optimizer updates with momentum-based gradient update components, thereby improving Spiking Transformer performance.

- We conduct experiments across multiple Spiking Transformer architectures using datasets from computer vision and natural language processing. The results show consistent performance improvements across different Spiking Transformers, demonstrating the effectiveness and generalizability of our method.

## 2. Related Work

### 2.1. Spiking Transformers

SNNs provide an efficient paradigm for next-generation machine intelligence, but their performance lags behind that of ANNs (Wei et al., 2024; Zhang et al., 2024; Liang et al., 2025; Wang et al., 2025a). To address this gap, Spiking Transformers combine SNNs with Transformer architectures, integrating energy efficiency with strong representation learning capabilities (Xiao et al., 2025). For example, Spikformer (Zhou et al., 2023b) pioneered this integration with Spiking Self Attention, achieving the first spike-driven computation in Transformers. Spike-driven Transformer (Yao et al., 2023) advanced this concept with a linear-complexity self-attention mechanism that significantly reduces energy requirements. This architecture was later expanded into Spike-driven Transformer V2 (Yao et al., 2024) to enhance versatility across vision tasks. QKformer (Zhou et al., 2024) introduced a novel spike-form Q-K attention mechanism with linear complexity, modeling token importance through binary vectors and achieving superior performance on ImageNet-1k. Spike-driven Transformer V3 (Yao et al., 2025) addressed inherent limitations in spiking neurons through Spike Firing Approximation, enabling integer training and spike-driven inference while improving accuracy and efficiency. For language modeling, SpikeLM (Xing et al., 2024) presented the first fully spiking mechanism for general language tasks with bi-directional and elastic spike encoding, narrowing the performance gap between SNNs and ANNs. Despite these advances, challenges remain in optimizing training processes due to the non-differentiability of spike functions, highlighting the need for specialized optimization algorithms for Spiking Transformers.

### 2.2. Adaptive Optimization Algorithms

Adaptive optimization algorithms have revolutionized deep learning by dynamically adjusting parameter-specific learning rates based on gradient history, enhancing convergence speed and model performance. These methods have become essential for training complex architectures like Spiking Transformers. Adam (Kingma & Ba, 2015) pioneered adaptive learning rates using first and second moment estimates, outperforming traditional methods in convergence speed. AdamW (Loshchilov & Hutter, 2019) improved generalization by decoupling weight decay from gradient updates. QHAdam (Ma & Yarats, 2019) introduces quasi-hyperbolic terms to replace Adam's moment estimators, allowing for better control of the update magnitude by decoupling momentum decay factors from immediate gradient contributions, resulting in improved training stability and performance. AdaX (Li et al., 2020) improves upon Adam by exponentially accumulating long-term gradient information to adaptively tune learning rates, addressing Adam's

tendency to converge prematurely to local minima in certain non-convex optimization landscapes. Lion (Chen et al., 2023) introduced a sign-based momentum approach that reduces memory usage compared to Adam while showing remarkable performance on large-scale tasks. While these optimizers excel in standard deep learning scenarios, non-differentiable spiking neurons present unique challenges. Our proposed AdaS algorithm specifically addresses these challenges by reconsidering the balance between adaptive updates and momentum-based gradient update components in surrogate gradient learning contexts.

## 3. Methodology

### 3.1. Quantitative Definition of Parameter-Update Noise

In gradient-based optimization, the ideal parameter update can be characterized by a proximal formulation (Nesterov, 2013). Given the current parameter vector $\boldsymbol{\theta}_{t-1}$, the step size $\eta_t$, and an objective function $f(\cdot)$, we define the ideal next-step parameter as

$$\boldsymbol{\theta}_t^\star = \arg\min_{\boldsymbol{\theta}} \frac{\|\boldsymbol{\theta} - \boldsymbol{\theta}_{t-1}\|^2}{2\eta_t} + f(\boldsymbol{\theta}). \quad (1)$$

The quadratic term constrains the update to a local neighborhood of $\boldsymbol{\theta}_{t-1}$, while the objective term encourages movement toward a loss-decreasing direction. Therefore, the ideal proximal update, i.e., $\boldsymbol{\theta}_t^\star - \boldsymbol{\theta}_{t-1}$, serves as a reference update for evaluating the actual optimizer update.

Let the actual parameter update at iteration $t$ be

$$\boldsymbol{r}_t = \boldsymbol{\theta}_t - \boldsymbol{\theta}_{t-1}. \quad (2)$$

We define the parameter-update noise as the increase in the objective value of Eq. (1) incurred by using $\boldsymbol{r}_t$ instead of the ideal proximal update.

**Definition 3.1. (Parameter-Update Noise).** Given an actual parameter update $\boldsymbol{r}_t$, the parameter-update noise is defined as

$$\mathcal{N}(\boldsymbol{r}_t) = \frac{\|\boldsymbol{r}_t\|^2}{2\eta_t} + f(\boldsymbol{\theta}_{t-1} + \boldsymbol{r}_t) - \frac{\|\boldsymbol{\theta}_t^\star - \boldsymbol{\theta}_{t-1}\|^2}{2\eta_t} - f(\boldsymbol{\theta}_t^\star). \quad (3)$$

Equivalently, by defining

$$h_t(\boldsymbol{\theta}) = \frac{\|\boldsymbol{\theta} - \boldsymbol{\theta}_{t-1}\|^2}{2\eta_t} + f(\boldsymbol{\theta}), \quad (4)$$

we have

$$\mathcal{N}(\boldsymbol{\theta}_t) = h_t(\boldsymbol{\theta}_t) - h_t(\boldsymbol{\theta}_t^\star). \quad (5)$$

Thus, $\mathcal{N}(\boldsymbol{\theta}_t)$ measures the deviation of the actual update from the ideal proximal update at iteration $t$.

In practice, directly computing $h_t(\boldsymbol{\theta}_t^\star)$ is intractable. To obtain a computable proxy, we replace the $h_t(\boldsymbol{\theta}_t^\star)$ with the proximal objective value at the current parameter $\boldsymbol{\theta}_{t-1}$, i.e., $h_t(\boldsymbol{\theta}_{t-1})$. This leads to the following estimated noise.

**Definition 3.2. (Estimated Parameter-Update Noise).** The estimated parameter-update noise is defined as

$$\mathcal{N}_{\text{est}}(\boldsymbol{\theta}_t) = \frac{\|\boldsymbol{\theta}_t - \boldsymbol{\theta}_{t-1}\|^2}{2\eta_t} + f(\boldsymbol{\theta}_t) - f(\boldsymbol{\theta}_{t-1}). \quad (6)$$

The estimated noise is a shifted proxy of the exact parameter-update noise. Specifically,

$$\mathcal{N}_{\text{est}}(\boldsymbol{\theta}_t) = \mathcal{N}(\boldsymbol{\theta}_t) + h_t(\boldsymbol{\theta}_t^\star) - h_t(\boldsymbol{\theta}_{t-1}). \quad (7)$$

For a fixed iteration $t$, the last two terms are independent of the candidate parameter update. Therefore, $\mathcal{N}_{\text{est}}$ preserves the relative comparison among parameter updates at the same iteration. Moreover, when the optimization trajectory converges to a local minimum $\boldsymbol{\theta}^\star$, both $\boldsymbol{\theta}_{t-1}$ and $\boldsymbol{\theta}_t^\star$ approach $\boldsymbol{\theta}^\star$, making the shift term asymptotically negligible.

We show that, to first order, the negative-gradient update minimizes this noise.

**Proposition 3.3.** *Under a first-order approximation of $f$ around $\boldsymbol{\theta}_{t-1}$, the parameter update that minimizes the first-order approximation of the parameter-update noise in Eq. (3) is*

$$\boldsymbol{r}_t^\star = -\eta_t \nabla f(\boldsymbol{\theta}_{t-1}). \quad (8)$$

*Therefore, the negative-gradient update is the locally noise-minimizing parameter update.*

### 3.2. Excessive Noise Problem in Spiking Transformers

In practical deep learning, models are optimized using mini-batches, while information over the full training set is often computationally prohibitive to obtain. Thus, the noise measured in our experiments is computed with respect to the objective actually used at each training step.

We denote the mini-batch objective at iteration $t$ by

$$f_t(\boldsymbol{\theta}) = \frac{1}{|B_t|} \sum_{(\boldsymbol{x}, \boldsymbol{y}) \in B_t} \ell(\boldsymbol{\theta}; \boldsymbol{x}, \boldsymbol{y}), \quad (9)$$

where $B_t$ is the sampled mini-batch and $\ell$ is the training loss used by the corresponding model. Given the actual update $\boldsymbol{r}_t$, the estimated parameter-update noise under the mini-batch objective is

$$\mathcal{N}_{\text{est},t} = \frac{\|\boldsymbol{r}_t\|^2}{2\eta_t} + f_t(\boldsymbol{\theta}_t) - f_t(\boldsymbol{\theta}_{t-1}). \quad (10)$$

**Assumption 3.4. (Comparable Mini-batch Stochasticity).** For both ANN and Spiking Transformers, the mini-batch gradient is an unbiased estimator of the gradient of the corresponding full-batch objective:

$$\mathbb{E}_{B_t}[\nabla f_t(\boldsymbol{\theta})] = \nabla F(\boldsymbol{\theta}). \quad (11)$$

Under identical training protocols, the variance induced by mini-batch sampling is assumed to be comparable across ANN and SNN training. Thus, mini-batch stochasticity is treated as a shared factor and is not regarded as the source of the additional noise specific to Spiking Transformers.

For ANN Transformers, the mini-batch objective is differentiable and can be directly optimized via backpropagation. In contrast, for Spiking Transformers, the optimizer cannot access the gradient of the original non-differentiable spiking objective and therefore relies on a surrogate objective. Let

$$\widetilde{f}_t(\boldsymbol{\theta}) = \frac{1}{|B_t|} \sum_{(\boldsymbol{x},\boldsymbol{y}) \in B_t} \widetilde{\ell}(\boldsymbol{\theta}; \boldsymbol{x}, \boldsymbol{y}) \qquad (12)$$

denote the surrogate mini-batch objective used for SNN training. According to Proposition 3.3[1], under a first-order approximation, the negative-gradient update minimizes the local parameter-update noise. Therefore, for the surrogate objective used in SNN training, the local noise-minimizing parameter update is

$$\widetilde{\boldsymbol{r}}_t^\star = -\eta_t \nabla \widetilde{f}_t(\boldsymbol{\theta}_{t-1}). \qquad (13)$$

Meanwhile, let

$$\boldsymbol{r}_{t,\mathrm{spk}}^\star = -\eta_t \boldsymbol{g}_t^{\mathrm{spk}} \qquad (14)$$

denote an ideal descent update associated with the original spiking objective, where $\boldsymbol{g}_t^{\mathrm{spk}}$ can be interpreted as a generalized gradient for the non-differentiable spiking dynamics. Then the deviation of the actual SNN update from the ideal spiking-objective descent update can be decomposed as

$$\boldsymbol{r}_t - \boldsymbol{r}_{t,\mathrm{spk}}^\star = \underbrace{\boldsymbol{r}_t - \widetilde{\boldsymbol{r}}_t^\star}_{\boldsymbol{n}_{\mathrm{opt}}} + \underbrace{\widetilde{\boldsymbol{r}}_t^\star - \boldsymbol{r}_{t,\mathrm{spk}}^\star}_{\boldsymbol{n}_{\mathrm{sur}}}. \qquad (15)$$

Here, $\boldsymbol{n}_{\mathrm{opt}}$ denotes the optimizer-induced deviation under the surrogate objective, while $\boldsymbol{n}_{\mathrm{sur}}$ denotes the discrepancy between the surrogate descent update and the ideal descent update of the original spiking objective.

Importantly, the measured SNN noise

$$\widetilde{\mathcal{N}}_{\mathrm{est},t} = \frac{\|\boldsymbol{r}_t\|^2}{2\eta_t} + \widetilde{f}_t(\boldsymbol{\theta}_t) - \widetilde{f}_t(\boldsymbol{\theta}_{t-1}) \qquad (16)$$

is computed only with respect to the surrogate objective. Under the first-order approximation, it satisfies

$$\widetilde{\mathcal{N}}_{\mathrm{est},t} \approx \frac{1}{2\eta_t} \|\boldsymbol{n}_{\mathrm{opt}}\|^2 - \frac{\eta_t}{2} \left\| \nabla \widetilde{f}_t(\boldsymbol{\theta}_{t-1}) \right\|^2. \qquad (17)$$

Thus, the measured SNN noise is a proxy for the optimizer-induced deviation within the surrogate-gradient training dynamics. It does not explicitly include the additional surrogate-induced discrepancy $\boldsymbol{n}_{\mathrm{sur}}$.

---

[1]The proof of Proposition 3.3 is provided in the Appendix A.1.

**Assumption 3.5. (Weak Deviation Correlation).** Given $\boldsymbol{\theta}_{t-1}$ and $B_t$, the optimizer-induced deviation $\boldsymbol{n}_{\mathrm{opt}}$ and the surrogate-induced discrepancy $\boldsymbol{n}_{\mathrm{sur}}$ are weakly correlated:

$$\left| \mathbb{E}\left[ \boldsymbol{n}_{\mathrm{opt}}^\top \boldsymbol{n}_{\mathrm{sur}} \mid \boldsymbol{\theta}_{t-1}, B_t \right] \right| \leq$$
$$\rho \left( \mathbb{E}\left[ \|\boldsymbol{n}_{\mathrm{opt}}\|^2 \mid \boldsymbol{\theta}_{t-1}, B_t \right] \mathbb{E}\left[ \|\boldsymbol{n}_{\mathrm{sur}}\|^2 \mid \boldsymbol{\theta}_{t-1}, B_t \right] \right)^{1/2},$$
$$(18)$$

where $\rho \ll 1$. This assumption is consistent with the concentration behavior of high-dimensional vectors, where weakly dependent directions tend to be nearly orthogonal (Vershynin, 2018).

**Proposition 3.6.** *Given the decomposition in Eq. (15), if the two deviation components $\boldsymbol{n}_{\mathrm{opt}}$ and $\boldsymbol{n}_{\mathrm{sur}}$ are orthogonal, then the total deviation from the ideal spiking-objective update satisfies*

$$\left\| \boldsymbol{r}_t - \boldsymbol{r}_{t,\mathrm{spk}}^\star \right\|^2 = \|\boldsymbol{n}_{\mathrm{opt}}\|^2 + \|\boldsymbol{n}_{\mathrm{sur}}\|^2. \qquad (19)$$

*Under Assumption 3.5, this relation approximately holds in expectation up to a small cross-correlation term.*

As discussed above, the measured noise $\widetilde{\mathcal{N}}_{\mathrm{est},t}$ mainly reflects the optimizer-induced deviation. Proposition 3.6 further shows that the surrogate-induced discrepancy contributes an additional component to the total deviation from the original spiking-objective update. Therefore, under Assumption 3.5, the actual parameter-update deviation in Spiking Transformers is expected to be no smaller than what is reflected by $\widetilde{\mathcal{N}}_{\mathrm{est},t}$, up to a small cross-correlation term.

As shown in Figure 1, Spiking Transformers exhibit substantially higher measured parameter-update noise than ANN Transformers. Since mini-batch stochasticity is shared and controlled by the experimental protocol, this observation indicates a stronger optimizer-induced parameter-update deviation under surrogate-gradient training. Furthermore, because the measured SNN noise is only a surrogate-level proxy, the actual parameter-update deviation with respect to the original spiking objective is expected to be more severe than what is directly observed. We refer to this phenomenon as the excessive noise problem.

### 3.3. AdaS: Adaptive Balancing of Update Components

Based on the analysis above, the excessive noise problem in Spiking Transformers can be mitigated by reducing the optimizer-induced deviation. Proposition 3.3 shows that, under a first-order approximation, the negative-gradient update is the locally noise-minimizing parameter update. This motivates incorporating gradient-update information into the update. However, directly following the negative gradient often leads to slow convergence, oscillations, instability, and difficulties in escaping saddle points or sharp local extrema. In contrast, momentum methods can alleviate these issues by accumulating historical gradient information (Ruder, 2016).

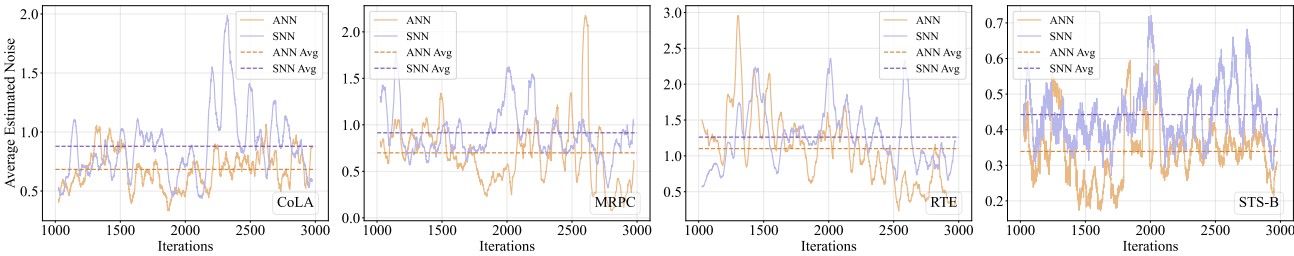

*Figure 1.* Comparison of average estimated parameter-update noise across layers between ANN Transformers and Spiking Transformers under the same mini-batch training settings.

Therefore, we propose AdaS, an adaptive optimizer that uses the first-order momentum as a stable approximation of the gradient-based update component and adaptively balances it with the adaptive optimizer update component. Specifically, AdaS updates the parameters as

$$\boldsymbol{\theta}_t = \boldsymbol{\theta}_{t-1} - \eta_t \left[ \alpha_t \boldsymbol{u}_t + (1 - \alpha_t) \boldsymbol{m}_t \right], \quad (20)$$

where $\boldsymbol{u}_t$ is the adaptive update component produced by the base optimizer, $\boldsymbol{m}_t$ is the first-order momentum, and $\alpha_t \in [0, 1]$ is an adaptive balancing coefficient.

Using the estimated noise in Eq. (6) and applying a first-order Taylor expansion of $f(\boldsymbol{\theta}_t)$ around $\boldsymbol{\theta}_{t-1}$, we have

$$\mathcal{N}_{\text{est}}(\boldsymbol{\theta}_t) \approx \frac{\|\boldsymbol{\theta}_t - \boldsymbol{\theta}_{t-1}\|^2}{2\eta_t} + \nabla f(\boldsymbol{\theta}_{t-1})^\top (\boldsymbol{\theta}_t - \boldsymbol{\theta}_{t-1}). \quad (21)$$

Substituting Eq. (20) into Eq. (21) gives

$$\mathcal{N}_{\text{est}}(\boldsymbol{\theta}_t) \approx \frac{\eta_t}{2} \|\alpha_t \boldsymbol{u}_t + (1 - \alpha_t) \boldsymbol{m}_t\|^2 \\ - \eta_t \nabla f(\boldsymbol{\theta}_{t-1})^\top \left[ \alpha_t \boldsymbol{u}_t + (1 - \alpha_t) \boldsymbol{m}_t \right]. \quad (22)$$

Since the first-order momentum is an exponential moving average of gradients, we use $\boldsymbol{m}_t \approx \nabla f(\boldsymbol{\theta}_{t-1})$ as a stable approximation of the current gradient vector. Then Eq. (22) can be simplified as

$$\mathcal{N}_{\text{est}}(\boldsymbol{\theta}_t) \approx \frac{\eta_t}{2} \left( \alpha_t^2 \|\boldsymbol{u}_t - \boldsymbol{m}_t\|^2 - \|\boldsymbol{m}_t\|^2 \right). \quad (23)$$

To maintain an appropriate noise level, AdaS encourages the estimated noise to approach a target value $\gamma$, where $\gamma \geq 0$. Accordingly, the balancing coefficient is obtained by solving

$$\alpha_t^\star = \arg\min_{\alpha_t} \left[ \frac{\eta_t}{2} \left( \alpha_t^2 \|\boldsymbol{u}_t - \boldsymbol{m}_t\|^2 - \|\boldsymbol{m}_t\|^2 \right) - \gamma \right]^2. \quad (24)$$

For $\alpha_t > 0$ and $\|\boldsymbol{u}_t - \boldsymbol{m}_t\| > 0$, the solution is

$$\alpha_t^\star = \frac{\sqrt{\|\boldsymbol{m}_t\|^2 + 2\gamma/\eta_t}}{\|\boldsymbol{u}_t - \boldsymbol{m}_t\|}. \quad (25)$$

Since $\alpha_t$ is a convex balancing coefficient, it is projected onto the interval $[0, 1]$:

$$\Pi_{[0,1]}(\varphi) = \arg\min_{\beta \in [0,1]} |\varphi - \beta|. \quad (26)$$

Thus, the final adaptive coefficient is

$$\alpha_t^\star = \Pi_{[0,1]} \left( \frac{\sqrt{\|\boldsymbol{m}_t\|^2 + 2\gamma/\eta_t}}{\|\boldsymbol{u}_t - \boldsymbol{m}_t\|} \right), \quad (27)$$

This adaptive balancing mechanism allows AdaS to effectively control the noise level in the parameter update while maintaining sufficient exploration. Finally, in Algorithm 1, we summarize the optimization procedure of our method, including the computation process of the adaptive weight $\alpha$.

---

**Algorithm 1** AdaS

---

**Require:** Initial parameters $\boldsymbol{\theta}_0$, learning rate $\{\eta_t\}_{t=1}^T$, weight decay factors $\{\lambda_t\}_{t=1}^T$, target noise level $\gamma$
**Ensure:** Optimized parameters $\boldsymbol{\theta}_T$
1:  **for** $t = 1, \ldots, T$ **do**
2:      Compute gradient $\boldsymbol{g}_t$
3:      Update the first-order moment $\boldsymbol{m}_t$
4:      Compute adaptive update component $\boldsymbol{u}_t$
5:      Compute balancing weight: $\alpha_t \leftarrow \frac{\sqrt{\|\boldsymbol{m}_t\|^2 + 2\gamma/\eta_t}}{\|\boldsymbol{u}_t - \boldsymbol{m}_t\| + \epsilon}$
6:      Project $\alpha_t$ onto $[0, 1]$: $\alpha_t \leftarrow \max(0, \min(1, \alpha_t))$
7:      $\boldsymbol{\theta}_t \leftarrow \boldsymbol{\theta}_{t-1} - \eta_t \left[ \alpha_t \boldsymbol{u}_t + (1 - \alpha_t) \boldsymbol{m}_t + \lambda_t \boldsymbol{\theta}_{t-1} \right]$
8:  **end for**

---

### 3.4. Convergence Analysis

We analyze the convergence of AdaS within the online learning framework established by (Zinkevich, 2003). In this framework, at each time step $t$, the optimization algorithm selects a parameter $\boldsymbol{\theta}_t \in \mathcal{X}$, where $\mathcal{X}$ is a feasible set, and its performance is evaluated by an unknown convex cost function $f_t(\cdot)$ at $\boldsymbol{\theta}_t$. Given that there exists an optimal parameter $\boldsymbol{\theta}^* = \arg\min_{\boldsymbol{\theta} \in \mathcal{X}} \sum_{t=1}^T f_t(\boldsymbol{\theta})$, we quantify the algorithm's performance using the regret:

$$\mathcal{R}(T) = \sum_{t=1}^T \left[ f_t(\boldsymbol{\theta}_t) - f_t(\boldsymbol{\theta}^*) \right]. \quad (28)$$

The regret serves as a metric for optimization efficiency, as smaller values of $f_t(\boldsymbol{\theta}_t) - f_t(\boldsymbol{\theta}^*)$ indicate that $\boldsymbol{\theta}_t$ is closer to the optimum $\boldsymbol{\theta}^*$. To ensure convergence to $\boldsymbol{\theta}^*$, it is

*Table 1.* Comparison of SpikeLM using AdaS with SpikeLM using other optimizers, as well as other ANNs and SNNs.

| Method | Optimizer | MNLI$_{m/mm}$ | QQP$_{F1}$ | QNLI | SST-2 | CoLA | STS-B | MRPC$_{F1}$ | RTE | Avg. |
|---|---|---|---|---|---|---|---|---|---|---|
| (Zhang et al., 2020) | AdamW | 47.2/47.3 | 67.0 | 61.3 | 80.6 | 0.0 | 4.7 | 81.2 | 52.7 | 49.1 |
| (Devlin et al., 2019) | – | 83.8/83.4 | 90.5 | 90.7 | 92.3 | 60.0 | 89.4 | 89.8 | 69.3 | 83.2 |
| (Xing et al., 2024) | AdamW | 56.8/55.2 | 70.0 | 60.6 | 80.6 | 14.6 | 20.0 | 82.3 | 53.8 | 54.9 |
| (Lv et al., 2025) | AdamW | 71.4/71.0 | 68.2 | 66.4 | 85.4 | 16.9 | 18.7 | 82.0 | 57.5 | 59.7 |
| (Zhou et al., 2026) | AdamW | 70.2/70.6 | 80.9 | 79.5 | 83.9 | 12.8 | 77.0 | 83.0 | 62.1 | 68.9 |
| | Lion | 76.6/77.0 | 84.4 | 84.3 | 86.0 | 37.0 | 85.4 | 84.8 | 69.3 | 76.1 |
| (Xing et al., 2024) | AdamW | 77.1/77.2 | 83.9 | 85.3 | 87.0 | 38.8 | 84.9 | 85.7 | 69.0 | 76.5 |
| | **AdaS** | 77.0/77.6 | 84.8 | 85.7 | 88.0 | 42.1 | 85.6 | 87.0 | 70.4 | 77.6 |

necessary that $\mathcal{R}(T)/T \to 0$ as $T \to \infty$. Without loss, we assume that AdaS is implemented based on Adam.

**Theorem 3.7.** *Assume that the function $f_t(\cdot)$ has bounded gradients, $\|\nabla f_t(\boldsymbol{\theta})\|_\infty \leq G_\infty$ for all $\boldsymbol{\theta} \in \mathbb{R}^d$ and distance between any $\boldsymbol{\theta_t}$ generated by AdaS is bounded, $\|\boldsymbol{\theta}_n - \boldsymbol{\theta}_m\|_\infty \leq D_\infty$ for any $m, n \in \{1, ..., T\}$, and $\beta_1, \beta_2 \in [0, 1)$ satisfy $\beta_1^2/\sqrt{\beta_2} \leq \sqrt{c} < 1$. Let $\eta_t = \eta/\sqrt{t}, \beta_{1,t} = \beta_1/\sqrt{t}$ and $\alpha_t = \alpha$. AdaS achieves the following guarantee, for all $T \geq 1$,*

$$\mathcal{R}(T) \leq \frac{dG_\infty D_\infty^2 \sqrt{T}}{2\eta \left[(1-\alpha)G_\infty + \alpha\right](1-\beta_1)} + \frac{2d\beta_1 G_\infty D_\infty \sqrt{T}}{1-\beta_1}$$
$$+ \frac{d(1-\alpha)\eta G_\infty^2 \sqrt{T}}{(1-\beta_1)^2} + \frac{d\alpha\eta G_\infty \sqrt{T}}{(1-\beta_1)^2(1-\beta_2)(1-c)}.$$
$$(29)$$

Next, we can demonstrate the regret of AdaS converges.

**Corollary 3.8.** *Assume that the function $f_t(\cdot)$ has bounded gradients, $\|\nabla f_t(\boldsymbol{\theta})\|_\infty \leq G_\infty$ for all $\boldsymbol{\theta} \in \mathbb{R}^d$ and distance between any $\boldsymbol{\theta_t}$ generated by AdaS is bounded, $\|\boldsymbol{\theta}_n - \boldsymbol{\theta}_m\|_\infty \leq D_\infty$ for any $m, n \in \{1, ..., T\}$. AdaS achieves the following guarantee, for all $T \geq 1$,*

$$\frac{\mathcal{R}(T)}{T} = \mathcal{O}(\frac{1}{\sqrt{T}}). \quad (30)$$

This result can be derived through the application of Theorem 3.7[2]. Thus, $\lim_{T \to \infty} \mathcal{R}(T)/T = 0$.

# 4. Experiment

## 4.1. Comparison With Other Optimizers

**Results for NLP tasks** Table 1 presents a comprehensive comparison of the SpikeLM (Xing et al., 2024) model using our proposed AdaS optimizer against various optimization methods and model architectures on the GLUE benchmark (Wang et al., 2018). When comparing SpikeLM

---

[2]The proof of Theorem 3.7 is provided in the Appendix A.2.

with AdaS against SpikeLM with AdamW, we observe consistent performance improvements across most tasks. This comparison is particularly important since our AdaS implementation is based on AdamW in this experiment, allowing us to directly assess the impact of our noise control mechanism. The average score across all GLUE tasks increased by 1.1%, indicating the general effectiveness of our proposed optimizer. Particularly notable improvements are observed on the CoLA task (+3.3%), RTE (+1.4%), and SST-2 (+1.0%). The only slight decrease occurred on MNLI-matched (-0.1%), which is negligible compared to the gains on other tasks, including MNLI-mismatched where performance improved by 0.4%. Compared to SpikeLM using the Lion optimizer, AdaS demonstrates more substantial improvements, with a 1.5% increase in the average score.

**Results for CV tasks** Table 2 and 3 presents comprehensive evaluations of our proposed AdaS optimizer against established optimization methods across three distinct CV tasks. We analyze the performance improvements achieved by the AdaS optimizer when training Spiking Transformer models compared to conventional optimization methods.

For the event-based tracking task, we evaluated the proposed AdaS optimizer using SDTrack-Tiny (Shan et al., 2025) on two representative benchmarks: FE108 (Zhang et al., 2021a) and VisEvent (Wang et al., 2023b). Experimental results demonstrate that SDTrack-Tiny trained with AdaS consistently outperforms its AdamW-trained counterpart on both datasets. Specifically, AdaS yields improvements of 1.2% in both Area Under the Curve (AUC) and Precision Rate (PR) on FE108, and achieves gains of 0.7% and 1.3% in AUC and PR on VisEvent, respectively. These results validate the generalization capability of AdaS across diverse visual tracking scenarios.

On the ADE20K dataset (Zhou et al., 2017), the E-SpikeFormer architecture (Yao et al., 2025) trained with AdaS achieves a mean Intersection over Union (mIoU) of 40.2%, representing a substantial improvement of 2.0 percentage points over the same model trained with AdamW

*Table 2.* Comparison of SDTrack with AdaS and AdamW optimizers, alongside other single object tracking pipelines, evaluated on two event-based tracking benchmarks FE108 (Zhang et al., 2021a) and VisEvent (Wang et al., 2023b).

| Methods | Optimizer | Param. (M) | Spiking Neuron | Timesteps $(T \times D)$ | FE108 | | VisEvent | |
|---|---|---|---|---|---|---|---|---|
| | | | | | AUC(%) | PR(%) | AUC(%) | PR(%) |
| (Yan et al., 2021) | - | 28.23 | - | $1 \times 1$ | 57.4 | 89.2 | 34.1 | 46.8 |
| (Wei et al., 2023) | - | 202.56 | - | $1 \times 1$ | 56.6 | 88.5 | 33.0 | 43.8 |
| (Kang et al., 2023) | - | 42.22 | - | $1 \times 1$ | 55.9 | 88.5 | 34.6 | 47.6 |
| (Zhang et al., 2022) | - | 20.55 | LIF | $3 \times 1$ | - | - | 35.0 | 50.3 |
| (Zhang et al., 2025a) | - | 31.40 | BA-LIF | $5 \times 1$ | - | - | 35.4 | 50.4 |
| (Shan et al., 2025) | AdamW | 19.61 | I-LIF | $1 \times 4$ | 59.0 | 91.3 | 35.6 | 49.2 |
| | **AdaS** | 19.61 | I-LIF | $1 \times 4$ | 60.2 | 92.5 | 36.3 | 50.5 |

*Table 3.* Comparison of Spiking Transformer trained with AdaS against alternative optimizers, alongside other ANN and SNN baselines, on the ADE20K and CIFAR10-DVS datasets. [†] Results are from our own implementation.

| Dataset | Type | Method | Architecture | Param(M) | Time | Optimizer | MIoU/Acc.(%) |
|---|---|---|---|---|---|---|---|
| ADE20K | ANN | (Yu et al., 2022) | ResNet-18 | 15.5 | – | SGD | 32.9 |
| | | (Wang et al., 2021) | PVT-Small | 28.2 | – | AdamW | 39.8 |
| | | (Wang et al., 2023a) | InternImage-T | 59.0 | – | AdamW | 48.1 |
| | SNN | (Yao et al., 2025) | E-SpikeFormer | 20.4 | 4 | SGD | 4.6[†] |
| | | | | 20.4 | 4 | Lion | 38.5[†] |
| | | | | 20.4 | 4 | AdamW | 38.2[†] |
| | | | | 20.4 | 4 | **AdaS** | 40.2 |
| CIFAR10-DVS | SNN | (Zhou et al., 2026) | Spikingformer-2-256 | 2.6 | 16 | AdamW | 81.3 |
| | | (Zhou et al., 2023a) | Spikformer-2-256 | 2.6 | 16 | AdamW | 80.9 |
| | | (Yao et al., 2023) | S-Transformer-2-256 | 2.6 | 16 | AdamW | 80.0 |
| | | (Zhou et al., 2024) | QKFormer-2-256 | 1.5 | 16 | SGD | 20.9[†] |
| | | | | 1.5 | 16 | Lion | 82.2[†] |
| | | | | 1.5 | 16 | AdamW | 84.0 |
| | | | | 1.5 | 16 | **AdaS** | 85.1 |

(38.2%). This performance gain is particularly significant considering the challenging nature of semantic segmentation tasks. Notably, AdaS also outperforms the Lion optimizer (38.5%) by 1.7 percentage points, demonstrating its superior capability in optimizing SNNs for dense prediction tasks. The poor performance of SGD (4.6%) on this dataset further emphasizes the necessity of adaptive optimization methods for training complex spiking transformer architectures.

For the neuromorphic CIFAR10-DVS dataset (Li et al., 2017), experiments with the QKFormer-2-256 architecture (Zhou et al., 2024) reveal that AdaS achieves an accuracy of 85.1%, surpassing AdamW by 1.1% (84.0%). This advantage demonstrates AdaS's enhanced capability to navigate the complex optimization landscape of neuromorphic vision tasks. AdaS also demonstrates superior performance compared to Lion (82.2%), with a margin of 2.9%.

### 4.2. Method Validation and Ablation Study

**Denoising Capability of AdaS** As shown in Figure 2, SpikeLM optimized with AdaS demonstrates lower noise levels during the optimization process compared to SpikeLM optimized with AdamW. On the CoLA and MRPC datasets, although the noise level of SpikeLM optimized with AdaS is slightly higher than that of BERT optimized with AdamW, the difference is minimal. Notably, on the RTE dataset, SpikeLM optimized with AdaS even exhibits lower noise levels than BERT optimized with AdamW. These experimental results validate two points: first, the AdaS optimizer effectively reduces noise during the optimization process by incorporating a first-order momentum mechanism. Second, properly reducing the noise in the optimization process of Spiking Transformers has a positive impact on improving the performance of SNN models.

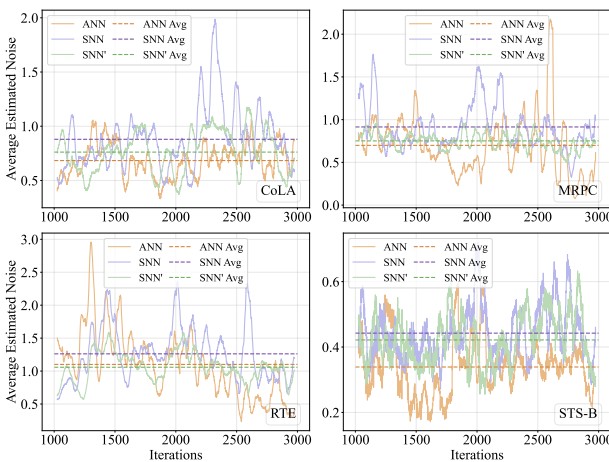

*Figure 2.* Comparison of average estimated noise across layers among BERT (ANN), SpikeLM (SNN) optimized with AdamW, and SpikeLM (SNN') optimized with AdaS.

**Effectiveness of Adaptive Balancing of Update Components** The experimental results presented in Table 4 demonstrate the effectiveness of adaptive balancing between update components in the AdaS optimizer across two distinct datasets: CoLA and CIFAR10-DVS. The comparison between AdaS with adaptive weights and its variant with fixed weights reveals consistent performance improvements. While AdaS with various fixed $\alpha$ values (0.9, 0.95, 0.99) shows some improvements over the baseline AdamW optimizer, these gains are inconsistent across datasets and $\alpha$ values. For instance, with $\alpha = 0.9$, AdaS improves CoLA performance by 2.1% but slightly decreases CIFAR10-DVS performance by 0.1%. In contrast, AdaS with adaptive weights demonstrates superior and more consistent improvements across both tasks. It achieves the highest accuracy on both datasets, with substantial gains of 3.3% on CoLA and 1.6% on CIFAR10-DVS. This indicates that dynamically adjusting the balance between update components is more effective than using any fixed weighting scheme.

*Table 4.* Comparison of AdaS with adaptive weights, AdaS with fixed weights, and AdamW, where $\alpha$ denotes the balance weight.
[†] Results are from our own implementation.

| Optimizer | $\alpha$ | Performance(%) | |
|---|---|---|---|
| | | **CoLA** | **CIFAR10-DVS** |
| AdamW | – | 38.8 | 83.5[†] |
| AdaS* | 0.9 | 40.9(+2.1) | 83.4(-0.1) |
| | 0.95 | 38.9(+0.1) | 84.2(+0.7) |
| | 0.99 | 39.8(+1.0) | 84.0(+0.5) |
| AdaS | Adaptive | 42.1(+3.3) | 85.1(+1.6) |

The adaptive weight $\alpha$ is designed to maintain noise within a reasonable range. Theoretically, AdaS with adaptive

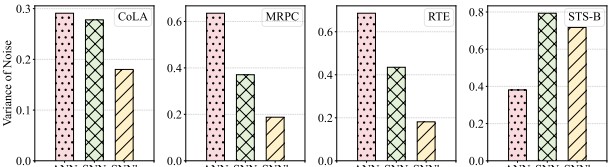

*Figure 3.* Comparison of variance of average estimated noise between BERT (ANN) and SpikeLM (SNN) optimized with AdamW, and SpikeLM (SNN') optimized with AdaS.

weighting should exhibit more stable noise distribution during optimization process. As shown in Figure 3, on the CoLA, MRPC, and RTE datasets, SpikeLM optimized with adaptive-weighted AdaS demonstrates lower noise variance during the optimization process than both SpikeLM optimized with AdamW and BERT optimized with AdamW. Similarly, on the STS-B dataset, SpikeLM optimized with adaptive-weighted AdaS also exhibits lower noise variance than SpikeLM optimized with AdamW. These experimental results confirm that the adaptive weight effectively stabilizes noise during the optimization process.

*Table 5.* Comparison of vanilla optimizers and their AdaS-augmented counterparts on the CoLA and CIFAR10-DVS datasets. +AdaS denotes the integration of the AdaS algorithm into the vanilla optimizer; Default refers to the vanilla optimizer alone.

| Optimizer | Setting | Performance(%) | |
|---|---|---|---|
| | | **CoLA** | **CIFAR10-DVS** |
| AdamW | Default | 38.8 | 83.5 |
| | +AdaS | 42.1(+3.3) | 85.1(+1.6) |
| AdaX | Default | 38.2 | 84.1 |
| | +AdaS | 40.5(+2.3) | 84.6(+0.5) |
| QHAdam | Default | 38.9 | 84.0 |
| | +AdaS | 40.4(+1.5) | 85.3(+1.3) |
| Adamax | Default | 39.2 | 81.2 |
| | +AdaS | 40.1(+0.9) | 81.9(+0.7) |

**Versatility of AdaS** Table 5 demonstrates AdaS's versatility across multiple optimization frameworks. When integrated with standard optimizers (AdamW, QHAdam, AdaX, and Adamax), AdaS consistently improved performance on both CoLA and CIFAR10-DVS datasets. Notably, AdamW exhibits the most substantial improvements when coupled with AdaS, achieving accuracy gains of 3.3 and 1.6 percentage points on CoLA and CIFAR10-DVS, respectively. Similarly, AdaX (+2.3/+0.5), QHAdam (+1.5/+1.3), and Adamax (+0.9/+0.7) all show meaningful performance increases across both datasets. This consistent pattern of improvement spanning different optimization algorithms and diverse dataset types provides compelling evidence for AdaS's versatility as a meta optimizer.

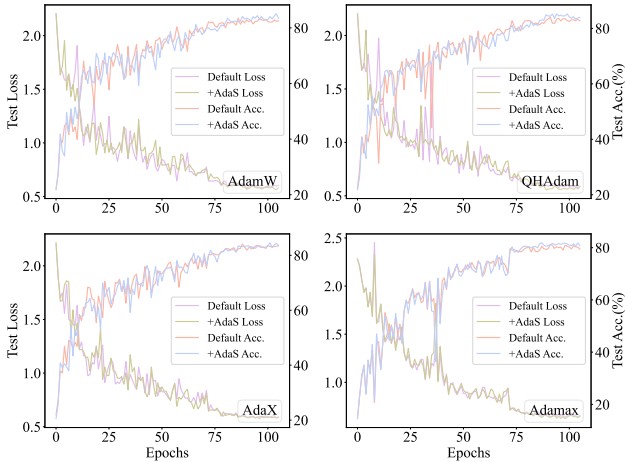

*Figure 4.* Comparison of test loss and accuracy curves for AdamW, QHAdam, AdaX, and Adamax optimizers with their AdaS algorithm variants on the CIFAR10-DVS dataset.

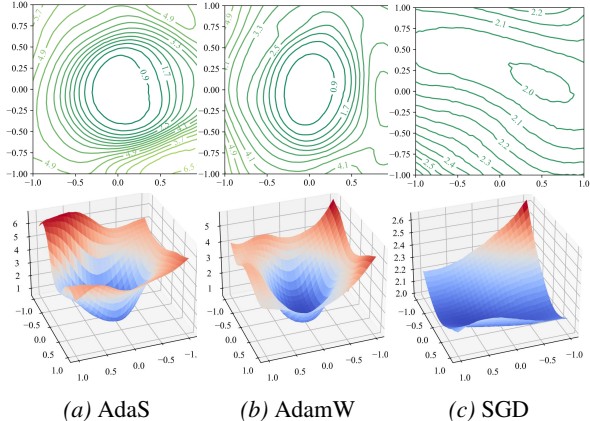

*(a)* AdaS     *(b)* AdamW     *(c)* SGD

*Figure 5.* Loss landscapes for QKformer optimized using AdaS, AdamW, and SGD on the CIFAR10-DVS dataset.

*Table 6.* Comparison of memory usage and training speed.

| Optimizer | Memory (MB) | Speed (iter./s) |
|---|---|---|
| AdamW | 10,672 | 3.51 |
| AdaS | 9,886 | 3.15 |

**Rate of Convergence** Figure 4 illustrates the test loss and accuracy curves of four optimizers—AdamW, QHAdam, AdaX, and Adamax—and their AdaS variants on the CIFAR10-DVS dataset. The results show that, for all four optimizers, the vanilla and AdaS-enhanced variants exhibit similar convergence rates. This finding indicates that integrating AdaS does not adversely affect the convergence speed of vanilla optimizers. In this experiment, the learning rates were set to 5e-3 for AdamW, QHAdam, and AdaX, while Adamax used a learning rate of 2e-4. The QKformer model was trained with a batch size of 16.

**Loss Landscape around Local Minima** As shown in Figure 5, we visualized the loss landscapes (Li et al., 2018) around local minima for QKformer optimized using AdaS, AdamW, and SGD on the CIFAR10-DVS dataset. Among these three optimizers, only SGD converged to a saddle point, indicating that introducing an appropriate level of noise during optimization indeed facilitates Spiking Transformers escape saddle points. The flatness of local minima reached by AdaS and AdamW appears comparable, suggesting that while AdaS reduces noise compared to AdamW, it does not compromise the model's generalization capability. In contrast, AdaS contributes to improved performance.

**Memory and Computational Overhead** We evaluate the practical overhead of AdaS on the CoLA task using an RTX 4090 GPU with a batch size of 16. As shown in Table 6, AdaS achieves comparable memory usage to AdamW, requiring 9,886 MB compared with 10,672 MB for AdamW. Meanwhile, AdaS obtains a training speed of 3.15 iter./s, compared with 3.51 iter./s for AdamW, corresponding to only about a 10% reduction in training speed. These results indicate that AdaS introduces only minor runtime overhead while maintaining a comparable memory footprint.

## 5. Conclusion

In this paper, we identified the excessive noise problem in training Spiking Transformers and analyzed its impact on optimization stability and model performance. To address this, we proposed AdaS, an adaptive optimization method that balances adaptive updates with a momentum-based gradient update component to maintain an appropriate noise level. Extensive experiments on computer vision and natural language processing tasks demonstrate that AdaS consistently improves the performance of various Spiking Transformers. Beyond these empirical gains, AdaS can be readily incorporated into existing adaptive optimization frameworks as a flexible meta-optimizer. Overall, these findings highlight the importance of optimization design for SNNs and provide an effective approach to narrowing the performance gap between SNNs and their ANN counterparts.

## Acknowledgements

This work was supported by the National Natural Science Foundation of China (Grants 62576080, 62220106008, and U2333211), by the Fundamental and Interdisciplinary Disciplines Breakthrough Plan of the Ministry of Education of China (JYB2025XDXM102), the Guangdong Introducing Innovative and Entrepreneurial Teams (Grant 2023ZT10×044), the Shenzhen Science and Technology Research Fund (Grant JCYJ20220818103001002), and the Project of Sichuan Engineering Technology Research Center for Civil Aviation Flight Technology and Flight Safety (No. GY2024-27D).

## Impact Statement

This paper presents work whose goal is to advance the field of Machine Learning. There are many potential societal consequences of our work, none of which we feel must be specifically highlighted here.

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

# A. Proofs of Propositions and Theorems

## A.1. Proof of Proposition 3.3.

**Proposition 3.3.** *Under a first-order approximation of $f$ around $\boldsymbol{\theta}_{t-1}$, the parameter update that minimizes the first-order approximation of the parameter-update noise in Eq. (3) is*

$$\boldsymbol{r}_t^\star = -\eta_t \nabla f(\boldsymbol{\theta}_{t-1}). \tag{31}$$

*Therefore, the negative gradient provides the locally noise-minimizing parameter update.*

*Proof.* Let

$$\boldsymbol{g}_t = \nabla f(\boldsymbol{\theta}_{t-1}) \tag{32}$$

denote the gradient of the objective at the current parameter. For an arbitrary parameter update

$$\boldsymbol{r} = \boldsymbol{\theta} - \boldsymbol{\theta}_{t-1}, \tag{33}$$

the parameter-update noise in Eq. (3) can be written, up to terms independent of $\boldsymbol{r}$, as

$$\mathcal{N}(\boldsymbol{r}) = \frac{\|\boldsymbol{r}\|^2}{2\eta_t} + f(\boldsymbol{\theta}_{t-1} + \boldsymbol{r}) + C_t, \tag{34}$$

where

$$C_t = -\frac{\|\boldsymbol{\theta}_t^\star - \boldsymbol{\theta}_{t-1}\|^2}{2\eta_t} - f(\boldsymbol{\theta}_t^\star) \tag{35}$$

is independent of the candidate parameter update $\boldsymbol{r}$.

Under the first-order Taylor approximation of $f$ around $\boldsymbol{\theta}_{t-1}$, we have

$$f(\boldsymbol{\theta}_{t-1} + \boldsymbol{r}) \approx f(\boldsymbol{\theta}_{t-1}) + \nabla f(\boldsymbol{\theta}_{t-1})^\top \boldsymbol{r}. \tag{36}$$

Substituting this approximation into $\mathcal{N}(\boldsymbol{r})$ yields the first-order approximate noise

$$\widehat{\mathcal{N}}(\boldsymbol{r}) = \frac{\|\boldsymbol{r}\|^2}{2\eta_t} + \boldsymbol{g}_t^\top \boldsymbol{r} + \widehat{C}_t, \tag{37}$$

where $\widehat{C}_t$ collects all terms independent of $\boldsymbol{r}$. Therefore, minimizing $\widehat{\mathcal{N}}(\boldsymbol{r})$ is equivalent to minimizing

$$q(\boldsymbol{r}) = \frac{\|\boldsymbol{r}\|^2}{2\eta_t} + \boldsymbol{g}_t^\top \boldsymbol{r}. \tag{38}$$

Since $\eta_t > 0$, $q(\boldsymbol{r})$ is a strictly convex quadratic function with Hessian

$$\nabla^2 q(\boldsymbol{r}) = \frac{1}{\eta_t} \boldsymbol{I} \succ 0. \tag{39}$$

Hence, $q$ has a unique global minimizer. Setting its gradient to zero gives

$$\nabla q(\boldsymbol{r}) = \frac{1}{\eta_t} \boldsymbol{r} + \boldsymbol{g}_t = \boldsymbol{0}. \tag{40}$$

Solving for $\boldsymbol{r}$, we obtain

$$\boldsymbol{r}_t^\star = -\eta_t \boldsymbol{g}_t = -\eta_t \nabla f(\boldsymbol{\theta}_{t-1}). \tag{41}$$

Equivalently, completing the square gives

$$q(\boldsymbol{r}) = \frac{1}{2\eta_t} \|\boldsymbol{r} + \eta_t \boldsymbol{g}_t\|^2 - \frac{\eta_t}{2} \|\boldsymbol{g}_t\|^2, \tag{42}$$

which is uniquely minimized when

$$\boldsymbol{r} = -\eta_t \boldsymbol{g}_t. \tag{43}$$

Thus, under the first-order approximation of the objective, the negative-gradient update yields the parameter update that minimizes the approximate parameter-update noise. This completes the proof. □

## A.2. Proof of Theorem 3.7.

**Theorem 3.7.** *Assume that the function $f_t(\cdot)$ has bounded gradients, $\|\nabla f_t(\boldsymbol{\theta})\|_\infty \leq G_\infty$, for all $\boldsymbol{\theta} \in \mathbb{R}^d$ and distance between any $\boldsymbol{\theta_t}$ generated by AdaS is bounded, $\|\boldsymbol{\theta}_n - \boldsymbol{\theta}_m\|_\infty \leq D_\infty$ for any $m, n \in \{1, ..., T\}$, and $\beta_1, \beta_2 \in [0, 1)$ satisfy $\beta_1^2/\sqrt{\beta_2} \leq \sqrt{c} < 1$. Let $\eta_t = \eta/\sqrt{t}, \beta_{1,t} = \beta_1/\sqrt{t}$ and $\alpha_t = \alpha$. AdaS achieves the following guarantee, for all $T \geq 1$,*

$$\mathcal{R}(T) \leq \frac{dG_\infty D_\infty^2 \sqrt{T}}{2\eta[(1-\alpha)G_\infty + \alpha](1-\beta_1)} + \frac{2d\beta_1 G_\infty D_\infty \sqrt{T}}{1-\beta_1} + \frac{d(1-\alpha)\eta G_\infty^2 \sqrt{T}}{(1-\beta_1)^2} + \frac{d\alpha\eta G_\infty \sqrt{T}}{(1-\beta_1)^2(1-\beta_2)(1-c)}. \tag{44}$$

*Proof.* The condition $\|\boldsymbol{\theta}_n - \boldsymbol{\theta}_m\|_\infty \leq D_\infty$ holds for any $m, n \in \{1, ..., T\}$, which implies that for any dimension $i$ of the variable: $|\theta_n^i - \theta_m^i| \leq D_\infty$ holds for any $m, n \in \{1, ..., T\}$. Let $\mathbf{g}_t$ denote $\nabla f_t(\boldsymbol{\theta})$. By our assumption, $\|\mathbf{g}_t\|_\infty \leq G_\infty$ for all $t$, or equivalently, $|g_{t,i}| \leq G_\infty$ for all $t$ and dimension $i$.

The regret is defined as:

$$\mathcal{R}(T) = \sum_{t=1}^{T}[f_t(\boldsymbol{\theta}_t) - f_t(\boldsymbol{\theta}^*)]. \tag{45}$$

Since $f_t(\boldsymbol{\theta})$ is a convex function, we have:

$$f_t(\boldsymbol{\theta}^*) \geq f_t(\boldsymbol{\theta}_t) + \langle \mathbf{g}_t, \boldsymbol{\theta}^* - \boldsymbol{\theta}_t \rangle. \tag{46}$$

Equivalently:

$$f_t(\boldsymbol{\theta}_t) - f_t(\boldsymbol{\theta}^*) \leq \langle \mathbf{g}_t, \boldsymbol{\theta}_t - \boldsymbol{\theta}^* \rangle. \tag{47}$$

Substituting into $\mathcal{R}(T)$:

$$\mathcal{R}(T) \leq \sum_{t=1}^{T} \langle \mathbf{g}_t, \boldsymbol{\theta}_t - \boldsymbol{\theta}^* \rangle = \sum_{i=1}^{d}\sum_{t=1}^{T} g_{t,i}(\theta_{t,i} - \theta_i^*). \tag{48}$$

The update process of AdaS is:

$$\begin{aligned}
\theta_{t+1,i} &= \theta_{t,i} - \eta_t(\frac{\alpha_t}{\sqrt{\hat{v}_{t,i}}} + 1 - \alpha_t)\hat{m}_{t,i} \\
&= \theta_{t,i} - \eta_t \frac{1}{1 - \prod_{s=1}^{t}\beta_{1,s}}(\frac{\alpha_t}{\sqrt{\hat{v}_{t,i}}} + 1 - \alpha_t)m_{t,i} \\
&= \theta_{t,i} - \eta_t \frac{1}{1 - \prod_{s=1}^{t}\beta_{1,s}}(\frac{\alpha_t}{\sqrt{\hat{v}_{t,i}}} + 1 - \alpha_t) \cdot [\beta_{1,t}m_{t-1,i} + (1 - \beta_{1,t})g_{t,i}].
\end{aligned} \tag{49}$$

Let $\xi_t = \frac{\eta_t}{1 - \prod_{s=1}^{t}\beta_{1,s}}$. Therefore:

$$\theta_{t+1,i} = \theta_{t,i} - \xi_t(\frac{\alpha_t}{\sqrt{\hat{v}_{t,i}}} + 1 - \alpha_t)[\beta_{1,t}m_{t-1,i} + (1 - \beta_{1,t})g_{t,i}]. \tag{50}$$

To isolate the term $g_{t,i}(\theta_{t,i} - \theta_i^*)$, we have:

$$\theta_{t+1,i} = \theta_{t,i} - \xi_t(\frac{\alpha_t}{\sqrt{\hat{v}_{t,i}}} + 1 - \alpha_t)[\beta_{1,t}m_{t-1,i} + (1 - \beta_{1,t})g_{t,i}]. \tag{51}$$

$\Longrightarrow$

$$(\theta_{t+1,i} - \theta_i^*)^2 = [(\theta_{t,i} - \theta_i^*) - \xi_t(\frac{\alpha_t}{\sqrt{\hat{v}_{t,i}}} + 1 - \alpha_t) \cdot [\beta_{1,t}m_{t-1,i} + (1 - \beta_{1,t})g_{t,i}]]^2. \tag{52}$$

$\Longrightarrow$

$$\begin{aligned}
&2\xi_t(\frac{\alpha_t}{\sqrt{\hat{v}_{t,i}}} + 1 - \alpha_t)[\beta_{1,t}m_{t-1,i} + (1 - \beta_{1,t})g_{t,i}](\theta_{t,i} - \theta_i^*) = \\
&(\theta_{t,i} - \theta_i^*)^2 - (\theta_{t+1,i} - \theta_i^*)^2 + \xi_t^2(\frac{\alpha_t}{\sqrt{\hat{v}_{t,i}}} + 1 - \alpha_t)^2 \cdot [\beta_{1,t}m_{t-1,i} + (1 - \beta_{1,t})g_{t,i}]^2.
\end{aligned} \tag{53}$$

$\Longrightarrow$

$$2\xi_t(\frac{\alpha_t}{\sqrt{\hat{v}_{t,i}}} + 1 - \alpha_t)(1 - \beta_{1,t})g_{t,i}(\theta_{t,i} - \theta_i^*) =$$

$$(\theta_{t,i} - \theta_i^*)^2 - (\theta_{t+1,i} - \theta_i^*)^2 - 2\xi_t(\frac{\alpha_t}{\sqrt{\hat{v}_{t,i}}} + 1 - \alpha_t)\beta_{1,t}m_{t-1,i}(\theta_{t,i} - \theta_i^*) + \xi_t^2(\frac{\alpha_t}{\sqrt{\hat{v}_{t,i}}} + 1 - \alpha_t)^2[\beta_{1,t}m_{t-1,i} + (1 - \beta_{1,t})g_{t,i}]^2.$$

$$(54)$$

$\Longrightarrow$

$$g_{t,i}(\theta_{t,i} - \theta_i^*) =$$

$$\frac{(\theta_{t,i} - \theta_i^*)^2 - (\theta_{t+1,i} - \theta_i^*)^2}{2\xi_t(\frac{\alpha_t}{\sqrt{\hat{v}_{t,i}}} + 1 - \alpha_t)(1 - \beta_{1,t})} - \frac{\beta_{1,t}m_{t-1,i}(\theta_{t,i} - \theta_i^*)}{1 - \beta_{1,t}} + \frac{\xi_t(\frac{\alpha_t}{\sqrt{\hat{v}_{t,i}}} + 1 - \alpha_t)[\beta_{1,t}m_{t-1,i} + (1 - \beta_{1,t})g_{t,i}]^2}{2(1 - \beta_{1,t})}. \qquad (55)$$

Given that $\beta_{1,t}m_{t-1,i} + (1 - \beta_{1,t})g_{t,i} = m_{t,i}$, we can decompose:

$$\sum_{i=1}^{d}\sum_{t=1}^{T} g_{t,i}(\theta_{t,i} - \theta_i^*) = \underbrace{\sum_{i=1}^{d}\sum_{t=1}^{T} \frac{(\theta_{t,i} - \theta_i^*)^2 - (\theta_{t+1,i} - \theta_i^*)^2}{2\xi_t(\frac{\alpha_t}{\sqrt{\hat{v}_{t,i}}} + 1 - \alpha_t)(1 - \beta_{1,t})}}_{(1)} - \underbrace{\sum_{i=1}^{d}\sum_{t=1}^{T} \frac{\beta_{1,t}m_{t-1,i}(\theta_{t,i} - \theta_i^*)}{1 - \beta_{1,t}}}_{(2)}$$

$$+ \underbrace{\sum_{i=1}^{d}\sum_{t=1}^{T} \frac{\xi_t(\frac{\alpha_t}{\sqrt{\hat{v}_{t,i}}} + 1 - \alpha_t)m_{t,i}^2}{2(1 - \beta_{1,t})}}_{(3)}. \qquad (56)$$

We will now derive upper bounds for terms (1), (2), and (3) individually.

**Bounding Term (1)** For term (1), we have:

$$\sum_{t=1}^{T} \frac{(\theta_{t,i} - \theta_i^*)^2 - (\theta_{t+1,i} - \theta_i^*)^2}{2\xi_t(\frac{\alpha_t}{\sqrt{\hat{v}_{t,i}}} + 1 - \alpha_t)(1 - \beta_{1,t})}$$

$$= \sum_{t=1}^{T} \frac{[(\theta_{t,i} - \theta_i^*)^2 - (\theta_{t+1,i} - \theta_i^*)^2](1 - \prod_{s=1}^{t}\beta_{1,s})}{2\eta_t(\frac{\alpha_t}{\sqrt{\hat{v}_{t,i}}} + 1 - \alpha_t)(1 - \beta_{1,t})} \qquad (57)$$

$$\leq \sum_{t=1}^{T} \frac{(\theta_{t,i} - \theta_i^*)^2 - (\theta_{t+1,i} - \theta_i^*)^2}{2\eta_t(\frac{\alpha_t}{\sqrt{\hat{v}_{t,i}}} + 1 - \alpha_t)(1 - \beta_{1,1})}.$$

Using telescoping series:

$$
\begin{aligned}
&\sum_{t=1}^{T} \frac{(\theta_{t,i} - \theta_i^*)^2 - (\theta_{t+1,i} - \theta_i^*)^2}{2\eta_t(\frac{\alpha_t}{\sqrt{\hat{v}_{t,i}}} + 1 - \alpha_t)(1 - \beta_{1,1})} \\
=&\sum_{t=1}^{T} \frac{(\theta_{t,i} - \theta_i^*)^2}{2\eta_t(\frac{\alpha_t}{\sqrt{\hat{v}_{t,i}}} + 1 - \alpha_t)(1 - \beta_{1,1})} - \sum_{t=1}^{T} \frac{(\theta_{t+1,i} - \theta_i^*)^2}{2\eta_t(\frac{\alpha_t}{\sqrt{\hat{v}_{t,i}}} + 1 - \alpha_t)(1 - \beta_{1,1})} \\
=&\frac{(\theta_{1,i} - \theta_i^*)^2}{2\eta_1(\frac{\alpha_1}{\sqrt{\hat{v}_{1,i}}} + 1 - \alpha_1)(1 - \beta_{1,1})} - \frac{(\theta_{T+1,i} - \theta_i^*)^2}{2\eta_T(\frac{\alpha_T}{\sqrt{\hat{v}_{T,i}}} + 1 - \alpha_T)(1 - \beta_{1,1})} \\
&+ \sum_{t=2}^{T} \frac{(\theta_{t,i} - \theta_i^*)^2}{1 - \beta_{1,1}}\Big[\frac{1}{2\eta_t(\frac{\alpha_t}{\sqrt{\hat{v}_{t,i}}} + 1 - \alpha_t)} - \frac{1}{2\eta_{t-1}(\frac{\alpha_{t-1}}{\sqrt{\hat{v}_{t-1,i}}} + 1 - \alpha_{t-1})}\Big] \\
\leq&\frac{(\theta_{1,i} - \theta_i^*)^2}{2\eta_1(\frac{\alpha_1}{\sqrt{\hat{v}_{1,i}}} + 1 - \alpha_1)(1 - \beta_{1,1})} - \frac{(\theta_{T+1,i} - \theta_i^*)^2}{2\eta_T(\frac{\alpha_T}{\sqrt{\hat{v}_{T,i}}} + 1 - \alpha_T)(1 - \beta_{1,1})} \\
&+ \frac{D_\infty^2}{1 - \beta_{1,1}} \sum_{t=2}^{T}\Big[\frac{1}{2\eta_t(\frac{\alpha_t}{\sqrt{\hat{v}_{t,i}}} + 1 - \alpha_t)} - \frac{1}{2\eta_{t-1}(\frac{\alpha_{t-1}}{\sqrt{\hat{v}_{t-1,i}}} + 1 - \alpha_{t-1})}\Big] \\
=&\frac{(\theta_{1,i} - \theta_i^*)^2}{2\eta_1(\frac{\alpha_1}{\sqrt{\hat{v}_{1,i}}} + 1 - \alpha_1)(1 - \beta_{1,1})} - \frac{(\theta_{T+1,i} - \theta_i^*)^2}{2\eta_T(\frac{\alpha_T}{\sqrt{\hat{v}_{T,i}}} + 1 - \alpha_T)(1 - \beta_{1,1})} \\
&+ \frac{D_\infty^2}{2\eta_T(\frac{\alpha_T}{\sqrt{\hat{v}_{T,i}}} + 1 - \alpha_T)(1 - \beta_{1,1})} - \frac{D_\infty^2}{2\eta_1(\frac{\alpha_1}{\sqrt{\hat{v}_{1,i}}} + 1 - \alpha_1)(1 - \beta_{1,1})}.
\end{aligned}
\tag{58}
$$

Since $-\frac{(\theta_{T+1,i} - \theta_i^*)^2}{2\eta_T(\frac{\alpha_T}{\sqrt{\hat{v}_{T,i}}} + 1 - \alpha_T)(1 - \beta_{1,1})} \leq 0$ and $\frac{(\theta_{1,i} - \theta_i^*)^2}{2\eta_1(\frac{\alpha_1}{\sqrt{\hat{v}_{1,i}}} + 1 - \alpha_1)(1 - \beta_{1,1})} \leq \frac{D_\infty^2}{2\eta_1(\frac{\alpha_1}{\sqrt{\hat{v}_{1,i}}} + 1 - \alpha_1)(1 - \beta_{1,1})}$, we have:

$$
\begin{aligned}
&\frac{(\theta_{1,i} - \theta_i^*)^2}{2\eta_1(\frac{\alpha_1}{\sqrt{\hat{v}_{1,i}}} + 1 - \alpha_1)(1 - \beta_{1,1})} - \frac{(\theta_{T+1,i} - \theta_i^*)^2}{2\eta_T(\frac{\alpha_T}{\sqrt{\hat{v}_{T,i}}} + 1 - \alpha_T)(1 - \beta_{1,1})} \\
&+ \frac{D_\infty^2}{2\eta_T(\frac{\alpha_T}{\sqrt{\hat{v}_{T,i}}} + 1 - \alpha_T)(1 - \beta_{1,1})} - \frac{D_\infty^2}{2\eta_1(\frac{\alpha_1}{\sqrt{\hat{v}_{1,i}}} + 1 - \alpha_1)(1 - \beta_{1,1})} \\
\leq&\frac{D_\infty^2}{2\eta_1(\frac{\alpha_1}{\sqrt{\hat{v}_{1,i}}} + 1 - \alpha_1)(1 - \beta_{1,1})} + \frac{D_\infty^2}{2\eta_T(\frac{\alpha_T}{\sqrt{\hat{v}_{T,i}}} + 1 - \alpha_T)(1 - \beta_{1,1})} - \frac{D_\infty^2}{2\eta_1(\frac{\alpha_1}{\sqrt{\hat{v}_{1,i}}} + 1 - \alpha_1)(1 - \beta_{1,1})} \\
=&\frac{D_\infty^2}{2\eta_T(\frac{\alpha_T}{\sqrt{\hat{v}_{T,i}}} + 1 - \alpha_T)(1 - \beta_{1,1})}.
\end{aligned}
\tag{59}
$$

Given that $v_{t,i} = (1 - \beta_2) \sum_{s=1}^{t} \beta_2^{t-s} g_{s,i}^2$. Therefore:

$$
\hat{v}_{t,i} = \frac{v_{t,i}}{1 - \beta_2^t} = \frac{(1 - \beta_2) \sum_{s=1}^{t} \beta_2^{t-s} g_{s,i}^2}{1 - \beta_2^t} \leq \frac{(1 - \beta_2) \sum_{s=1}^{t} \beta_2^{t-s} G_\infty^2}{1 - \beta_2^t} = \frac{(1 - \beta_2^t) G_\infty^2}{1 - \beta_2^t} = G_\infty^2.
\tag{60}
$$

Since $\eta_T = \frac{\eta}{\sqrt{T}}, \alpha_T = \alpha$ and $\beta_{1,1} = \beta_1$, we obtain:

$$
\sum_{i=1}^{d} \frac{D_\infty^2}{2\eta_T(\frac{\alpha_T}{\sqrt{\hat{v}_{T,i}}} + 1 - \alpha_T)(1 - \beta_{1,1})} \leq \frac{d G_\infty D_\infty^2 \sqrt{T}}{2\eta[(1 - \alpha)G_\infty + \alpha](1 - \beta_1)}.
\tag{61}
$$

**Bounding Term (2)**    Given that $m_{t,i} = \sum_{s=1}^{t}(1 - \beta_{1,s})(\prod_{k=s+1}^{t} \beta_{1,k}) \cdot g_{s,i}$, we have:

$$
|m_{t,i}| \leq \sum_{s=1}^{t}(1 - \beta_{1,s})(\prod_{k=s+1}^{t} \beta_{1,k})|g_{s,i}| \leq G_\infty \sum_{s=1}^{t}(1 - \beta_{1,s})(\prod_{k=s+1}^{t} \beta_{1,k}) = G_\infty(1 - \prod_{s=1}^{t} \beta_{1,s}) \leq G_\infty.
\tag{62}
$$

Therefore:

$$-\sum_{i=1}^{d}\sum_{t=1}^{T}\frac{\beta_{1,t}m_{t-1,i}(\theta_{t,i}-\theta_i^*)}{1-\beta_{1,t}}$$

$$=\sum_{i=1}^{d}\sum_{t=1}^{T}\frac{\beta_{1,t}m_{t-1,i}(\theta_i^*-\theta_{t,i})}{1-\beta_{1,t}}$$

$$\leq\sum_{i=1}^{d}\sum_{t=1}^{T}\frac{\beta_{1,t}|m_{t-1,i}||\theta_i^*-\theta_{t,i}|}{1-\beta_{1,t}}$$

$$\leq\frac{dG_\infty D_\infty\sum_{t=1}^{T}\beta_{1,t}}{1-\beta_1}. \tag{63}$$

Furthermore, since

$$\sum_{t=1}^{T}\beta_{1,t}=\sum_{t=1}^{T}\frac{\beta_1}{\sqrt{t}}\leq\beta_1(1+\int_1^T\frac{dt}{\sqrt{t}})=\beta_1(2\sqrt{T}-1)\leq 2\beta_1\sqrt{T}, \tag{64}$$

we have:

$$\frac{dG_\infty D_\infty\sum_{t=1}^{T}\beta_{1,t}}{1-\beta_1}\leq\frac{2d\beta_1 G_\infty D_\infty\sqrt{T}}{1-\beta_1}. \tag{65}$$

**Bounding Term (3)**

$$\sum_{i=1}^{d}\sum_{t=1}^{T}\frac{\xi_t(\frac{\alpha_t}{\sqrt{\hat{v}_{t,i}}}+1-\alpha_t)m_{t,i}^2}{2(1-\beta_{1,t})}=\sum_{i=1}^{d}\sum_{t=1}^{T}\frac{\xi_t\alpha_t m_{t,i}^2}{2\sqrt{\hat{v}_{t,i}}(1-\beta_{1,t})}+\sum_{i=1}^{d}\sum_{t=1}^{T}\frac{\xi_t(1-\alpha_t)m_{t,i}^2}{2(1-\beta_{1,t})}. \tag{66}$$

For the second component:

$$\sum_{i=1}^{d}\sum_{t=1}^{T}\frac{\xi_t(1-\alpha_t)m_{t,i}^2}{2(1-\beta_{1,t})}=\sum_{i=1}^{d}\sum_{t=1}^{T}\frac{\eta_t(1-\alpha_t)m_{t,i}^2}{2(1-\beta_{1,t})(1-\prod_{s=1}^{t}\beta_{1,s})}\leq\frac{d(1-\alpha)G_\infty^2\sum_{t=1}^{T}\eta_t}{2(1-\beta_1)^2}\leq\frac{d(1-\alpha)\eta G_\infty^2\sqrt{T}}{(1-\beta_1)^2}. \tag{67}$$

For the first component, transforming $m_{t,i}^2$ and applying the Cauchy-Schwarz inequality:

$$m_{t,i}^2=[\sum_{s=1}^{t}\frac{(1-\beta_{1,s})(\prod_{r=s+1}^{t}\beta_{1,r})}{\sqrt{(1-\beta_2)\beta_2^{t-s}}}\sqrt{(1-\beta_2)\beta_2^{t-s}}g_{s,i}]^2$$

$$\leq\sum_{s=1}^{t}[\frac{(1-\beta_{1,s})(\prod_{r=s+1}^{t}\beta_{1,r})}{\sqrt{(1-\beta_2)\beta_2^{t-s}}}]^2\cdot\sum_{s=1}^{t}[\sqrt{(1-\beta_2)\beta_2^{t-s}}g_{s,i}]^2$$

$$=\sum_{s=1}^{t}\frac{(1-\beta_{1,s})^2(\prod_{r=s+1}^{t}\beta_{1,r})^2}{(1-\beta_2)\beta_2^{t-s}}\underbrace{\sum_{s=1}^{t}(1-\beta_2)\beta_2^{t-s}g_{s,i}^2}_{v_{t,i}}. \tag{68}$$

Therefore:

$$\frac{\xi_t\alpha_t m_{t,i}^2}{2\sqrt{\hat{v}_{t,i}}(1-\beta_{1,t})}$$

$$\leq\frac{\xi_t\alpha_t}{2\sqrt{v_{t,i}}(1-\beta_{1,t})}\sum_{s=1}^{t}\frac{(1-\beta_{1,s})^2(\prod_{r=s+1}^{t}\beta_{1,r})^2}{(1-\beta_2)\beta_2^{t-s}}v_{t,i}$$

$$=\frac{\xi_t\alpha_t\sqrt{v_{t,i}}}{2(1-\beta_{1,t})}\sum_{s=1}^{t}\frac{(1-\beta_{1,s})^2(\prod_{r=s+1}^{t}\beta_{1,r})^2}{(1-\beta_2)\beta_2^{t-s}}$$

$$\leq\frac{\xi_t\alpha_t G_\infty}{2(1-\beta_{1,t})}\sum_{s=1}^{t}\frac{(1-\beta_{1,s})^2(\prod_{r=s+1}^{t}\beta_{1,r})^2}{(1-\beta_2)\beta_2^{t-s}}. \tag{69}$$

Considering:

$$\frac{\xi_t}{2(1-\beta_{1,t})} \sum_{s=1}^{t} \frac{(1-\beta_{1,s})^2 (\prod_{r=s+1}^{t} \beta_{1,r})^2}{(1-\beta_2)\beta_2^{t-s}}$$

$$= \frac{\eta_t}{2(1-\beta_{1,t})(1-\prod_{s=1}^{t}\beta_{1,s})} \cdot \sum_{s=1}^{t} (\frac{1-\beta_{1,s}}{\sqrt{1-\beta_2}})^2 \prod_{r=s+1}^{t} (\frac{\beta_{1,r}}{\sqrt{\beta_2}})^2. \tag{70}$$

Under the assumption that $\frac{\beta_{1,t}}{\sqrt{\beta_2}} \leq \sqrt{c} < 1$ and using the geometric series formula:

$$\frac{\eta_t}{2(1-\beta_{1,t})(1-\prod_{s=1}^{t}\beta_{1,s})} \cdot \sum_{s=1}^{t} (\frac{1-\beta_{1,s}}{\sqrt{1-\beta_2}})^2 \prod_{r=s+1}^{t} (\frac{\beta_{1,r}}{\sqrt{\beta_2}})^2$$

$$\leq \frac{\eta_t}{2(1-\beta_1)^2(1-\beta_2)} \sum_{s=1}^{t} \prod_{r=s+1}^{t} c \leq \frac{\eta_t}{2(1-\beta_1)^2(1-\beta_2)(1-c)}. \tag{71}$$

Further developing this inequality:

$$\sum_{i=1}^{d} \sum_{t=1}^{T} \frac{\xi_t \alpha_t G_\infty}{2(1-\beta_{1,t})} \sum_{s=1}^{t} \frac{(1-\beta_{1,s})^2 (\prod_{r=s+1}^{t} \beta_{1,r})^2}{(1-\beta_2)\beta_2^{t-s}}$$

$$\leq \frac{d\alpha G_\infty \sum_{t=1}^{T} \eta_t}{2(1-\beta_1)^2(1-\beta_2)(1-c)} \leq \frac{d\alpha\eta G_\infty \sqrt{T}}{(1-\beta_1)^2(1-\beta_2)(1-c)}. \tag{72}$$

Consolidating all the bounds derived for terms (1), (2), and (3), we establish the following upper bound for the regret:

$$\mathcal{R}(T) \leq \frac{dG_\infty D_\infty^2 \sqrt{T}}{2\eta[(1-\alpha)G_\infty + \alpha](1-\beta_1)} + \frac{2d\beta_1 G_\infty D_\infty \sqrt{T}}{1-\beta_1} + \frac{d(1-\alpha)\eta G_\infty^2 \sqrt{T}}{(1-\beta_1)^2} + \frac{d\alpha\eta G_\infty \sqrt{T}}{(1-\beta_1)^2(1-\beta_2)(1-c)}. \tag{73}$$

This completes the proof of Theorem 3.7. $\qquad\square$

# B. Additional Experimental Results

**Results for CIFAR-100 Dataset** The CIFAR-100 dataset (Krizhevsky et al., 2009) is a widely-used benchmark for image classification tasks in computer vision research. This dataset comprises 60,000 32×32 color images distributed across 100 fine-grained classes, with 600 images per class. The dataset is partitioned into 50,000 training images and 10,000 test images, maintaining a balanced distribution across all categories.

On the CIFAR-100 classification dataset, we employed a batch size of 64 and optimizer-specific learning rates: $3 \times 10^{-4}$ for AdaS and AdamW, $3 \times 10^{-5}$ for Lion, and $3 \times 10^{-3}$ for SGD. The value of $\gamma$ is 2.0. The S-Transformer-2-512 model optimized with AdaS achieves 79.1% accuracy, outperforming the AdamW-trained counterpart by 0.7% (78.4%). AdaS also demonstrates superior performance over Lion (78.0%) with a 1.1% improvement.

*Table 7.* Comparison between Spiking Transformer using AdaS and the same model using other optimizers, as well as other ANNs and SNNs on CIFAR-100 dataset. [†] Results are from our own implementation.

| Type | Method | Optimizer | Acc.(%) |
|------|--------|-----------|---------|
| ANN | ResNet-19 | SGD | 75.4 |
| | Transformer-4-384 | AdamW | 81.0 |
| SNN | (Li et al., 2021) | SGD | 74.2 |
| | (Meng et al., 2022) | SGD | 78.5 |
| | (Zhou et al., 2023b) | AdamW | 78.2 |
| | (Yao et al., 2023) | SGD | 56.1[†] |
| | | Lion | 78.0[†] |
| | | AdamW | 78.4 |
| | | **AdaS** | 79.1 |

**Energy Analysis of AdaS** We analyze whether AdaS affects the energy consumption of SNN models. Following the commonly used 45nm technology node estimates, we set the energy cost of an accumulate operation and a multiply-accumulate operation to $E_{\mathrm{AC}} = 0.9$ pJ and $E_{\mathrm{MAC}} = 4.6$ pJ, respectively (Horowitz, 2014). The total energy consumption is computed as

$$E_{\mathrm{total}} = N_{\mathrm{AC}}E_{\mathrm{AC}} + N_{\mathrm{MAC}}E_{\mathrm{MAC}}, \tag{74}$$

where $N_{\mathrm{AC}}$ and $N_{\mathrm{MAC}}$ denote the numbers of AC and MAC operations, respectively. Specifically, AC operations are estimated from the non-zero spike activations in spike-driven modules, while MAC operations mainly include LIF state updates and the remaining dense computations.

For SpikeLM on the GLUE benchmark, the model trained with AdaS consumes 13.72 mJ, which is nearly identical to the 13.74 mJ consumed by the AdamW-trained counterpart. Similarly, on CIFAR10-DVS with QKFormer-2-256, both AdamW and AdaS result in an energy consumption of 9.17 mJ. On ADE20K with E-SpikeFormer, AdaS consumes 74.25 mJ compared with 72.95 mJ for AdamW, showing only a marginal difference. Since AdaS only modifies the training optimization process and does not introduce additional inference operations, these results demonstrate that it has a negligible impact on inference-stage energy consumption, while improving training effectiveness without compromising the energy efficiency of SNN models during inference.

## C. Experimental Details

All experiments are conducted on a server equipped with 8 RTX 4090 GPUs. The server's CPU is an Intel(R) Xeon(R) Gold 6348 CPU @ 2.60GHz, and the operating system is Ubuntu 22.04.

### C.1. GLUE

The General Language Understanding Evaluation (GLUE) (Wang et al., 2018) benchmark serves as a standardized evaluation framework for natural language understanding (NLU) systems. It consists of a collection of diverse tasks covering various linguistic phenomena, including linguistic acceptability, sentiment analysis, paraphrase detection, semantic textual similarity, natural language inference, and question answering. In this work, we report results on eight GLUE tasks: Corpus of Linguistic Acceptability (CoLA) (Warstadt et al., 2019), Stanford Sentiment Treebank (SST-2) (Socher et al., 2013), Microsoft Research Paraphrase Corpus (MRPC) (Dolan & Brockett, 2005), Semantic Textual Similarity Benchmark (STS-B) (Cer et al., 2017), Quora Question Pairs (QQP), Multi-Genre Natural Language Inference (MNLI) (Williams et al., 2018), Question Natural Language Inference (QNLI) (Rajpurkar et al., 2016), and Recognizing Textual Entailment (RTE) (Dagan et al., 2005; Haim et al., 2006; Giampiccolo et al., 2007; Bentivogli et al., 2009).

In the experiment on GLUE, we employ a batch size of 16. For the AdaS and AdamW optimizers, the learning rate is set to $2 \times 10^{-5}$. Whereas for the Lion optimizer, since its update magnitude has an absolute value consistently equal to 1—typically requiring a learning rate approximately 10 times smaller than AdamW to achieve comparable update magnitudes—the learning rate is set to $2 \times 10^{-6}$. For MRPC and STS-B dataset, the AdaS hyperparameter $\gamma$ is set to 1.5, while for the other datasets it is set to 2.0. We report F1 scores for QQP and MRPC, Spearman correlation for STS-B, Matthews correlation coefficient for CoLA, and accuracy for the remaining tasks.

### C.2. FE108 and VisEvent

FE108 (Zhang et al., 2021a) is a large-scale dataset for single object tracking that contains 108 video sequences totaling 1.5 hours, with annotations provided at 40Hz for frames and 240Hz for events. The dataset was captured using a DAVIS346 camera that simultaneously records both conventional grayscale images and event data, with precise ground truth bounding boxes provided by a motion capture system. It includes 21 different object types and covers challenging conditions such as low-light environments, high dynamic range scenes, and fast-moving objects. FE108 represents the largest available dataset combining frame and event data for object tracking research.

The VisEvent dataset (Wang et al., 2023b) is a large-scale benchmark for object tracking that combines RGB and event cameras. It contains 820 video pairs with 371,128 manually annotated frames, divided into 500 training and 320 testing videos. The dataset captures challenging scenarios including low illumination, high-speed motion, and background clutter. VisEvent leverages the complementary strengths of both camera types: RGB cameras capture texture details effectively, while event cameras provide blur-free imaging under fast motion and low-light conditions. This dataset has facilitated the development of over 30 dual-modality tracking methods and serves as a benchmark for evaluating multi-modal tracking algorithms.

The experimental setup largely follows that of (Shan et al., 2025). It should be particularly noted that on the FE108 dataset, the weight decay is set to 0.003 to mitigate overfitting. On both datasets, $\gamma$ is set to 1.0.

## C.3. ADE20K

The ADE20K dataset (Zhou et al., 2017) is a comprehensive scene parsing dataset comprising over 20,000 densely annotated images for semantic segmentation tasks. This dataset encompasses a diverse collection of indoor and outdoor scenes with pixel-level annotations across more than 150 semantic categories, including both objects and scene elements. The dataset's extensive coverage of real-world scenarios has established it as a standard benchmark for semantic segmentation models.

For experiments conducted on this dataset, we employed a batch size of 12 across all training configurations. The learning rate was carefully tuned for each optimization algorithm: AdaS and AdamW were both configured with a learning rate of $1 \times 10^{-3}$, while Lion utilized a more conservative learning rate of $1 \times 10^{-4}$, and SGD employed a higher learning rate of $1 \times 10^{-2}$. The hyperparameter $\gamma$ was set to 1.0 for all experimental conditions to ensure consistent comparison across different optimizers.

## C.4. CIFAR10-DVS

CIFAR10-DVS (Li et al., 2017) is a neuromorphic dataset derived from the widely-used CIFAR-10 benchmark, specifically designed for event-driven vision applications. This dataset converts the original static RGB images of CIFAR-10 into spike-based representations captured by Dynamic Vision Sensors (DVS). The dataset comprises 10,000 samples across 10 object categories, with each sample consisting of asynchronous event streams that encode temporal changes in pixel intensity rather than conventional frame-based imagery.

For our experiments on this dataset, we adopted a batch size of 16. We configured each optimizer with algorithm-specific learning rates that yielded optimal convergence characteristics: AdaS and AdamW were both set to $5 \times 10^{-3}$, Lion employed a reduced learning rate of $5 \times 10^{-4}$ to ensure stable training dynamics, while SGD required a higher learning rate of $5 \times 10^{-2}$ to achieve comparable performance. The hyperparameter $\gamma$ was consistently maintained at 1.0 across all experimental conditions.

# D. Evaluation Metrics

### D.1. Mean Intersection over Union (MIoU)

Mean Intersection over Union is a standard evaluation metric for semantic segmentation tasks. The IoU for a single class is calculated as the ratio of the intersection area between the predicted segmentation and ground truth to their union area. Mathematically, $\text{IoU} = |A \cap B|/|A \cup B|$, where $A$ represents the predicted pixels and $B$ represents the ground truth pixels for a specific class. MIoU is computed by averaging the IoU scores across all classes in the dataset. This metric ranges from 0 to 1, with higher values indicating better segmentation performance. MIoU is particularly valuable because it penalizes both false positives and false negatives, providing a comprehensive assessment of segmentation quality.

### D.2. Area Under the Curve (AUC)

The Area Under the Curve (AUC) is a single number that measures how well a tracking algorithm performs overall. It is calculated by measuring the area underneath the Success Plot curve. In the Success Plot, the x-axis shows different precision thresholds (from 0 to 1), and the y-axis shows the success rate (percentage of frames that meet each threshold). The AUC takes all these success rates and combines them into one score. A higher AUC score means better tracking performance. The maximum possible AUC is 1, which would indicate perfect tracking. The minimum is 0, indicating complete failure. The main advantage of AUC is that it gives researchers a simple way to compare different tracking methods with just one number, rather than having to look at the entire curve. This makes it easier to determine which algorithm works best overall across different conditions.

### D.3. Precision Rate (PR)

Precision Rate, commonly referred to as Precision, measures the proportion of true positive predictions among all positive predictions made by the model. It is calculated as $\text{Precision} = \text{TP}/(\text{TP} + \text{FP})$, where TP represents true positives and FP represents false positives. This metric is particularly important in applications where false positives carry significant costs. In this study, PR represents the percentage of samples where the distance between the predicted and ground truth target centers is below 20 pixels.

