# OpenReview forum: "AdaS: Adaptive Gradient Descent for Spiking Transformers"
_ICML.cc/2026/Conference — ICML 2026 regular_

### Official Review · Reviewer_NW7G · 2026-03-09

**Soundness:** 3
**Presentation:** 3
**Significance:** 3
**Originality:** 2
**Overall Recommendation:** 3
**Confidence:** 4

**Summary:**

The authors quantitatively define update-direction noise in spiking neural networks (SNNs) and point out that the superposition of noise from surrogate gradients and adaptive optimizers is the key reason why SNN training is unstable and often underperforms traditional artificial neural networks (ANNs). Based on this insight, AdaS adaptively balances the update direction of adaptive optimizers with the first-order momentum (which approximates the steepest descent direction), thereby controlling the training noise at an optimal level. This approach ensures stable convergence without sacrificing generalization ability. From a theoretical perspective, the authors provide a formal definition of update-direction noise, derive the mechanism of noise superposition, and establish a regret convergence proof for AdaS. Empirically, they validate AdaS across multiple tasks—including NLP (GLUE) and computer vision tasks (CIFAR10-DVS, ADE20K, and event-based tracking)—and across various spiking Transformer architectures, demonstrating consistent improvements over mainstream optimizers such as AdamW and Lion.

**Compliance With Llm Reviewing Policy:**

Affirmed.

**Final Justification:**

Overall, while the authors have provided a detailed response to the review comments and supplemented hyperparameter sensitivity experiments and model scale justifications, the core limitations remain insufficiently addressed, including marginal performance gains on some tasks, lack of systematic validation on larger spiking models, and incomplete justification of theoretical generality.We therefore uphold the original "weak reject" recommendation and advise the authors to revise the manuscript by addressing the above issues before resubmission.

**Key Questions For Authors:**

1. The hyperparameter γ takes different values across tasks (e.g., GLUE, CIFAR10-DVS, ADE20K). Have the authors conducted systematic sensitivity experiments on γ? Does the choice of γ significantly affect the stability and final performance of AdaS?

2. The paper proves the regret convergence of AdaS when built upon Adam. Can this result be directly extended to other adaptive optimizers such as QHAdam, AdaX, or Adamax? Are there any theoretical limitations to such extensions?

3. The experiments mainly focus on medium-scale models. Would AdaS remain effective on deeper and larger spiking Transformers or large-scale pretrained SpikeLM models? Are there any potential training stability issues in those settings?

**Limitations:**

Yes.

**Strengths And Weaknesses:**

Strengths

The paper proposes the AdaS framework, which adaptively balances the steepest descent direction and adaptive updates, rather than simply modifying existing optimizers. It provides a quantitative definition of update-direction noise, proves that the steepest descent direction can minimize such noise, and presents a regret convergence analysis of AdaS under the online learning framework. Moreover, AdaS can be directly integrated into existing spiking Transformer architectures, improving accuracy while preserving the low-power advantages of SNNs.

Weaknesses

The selection of the hyperparameter γ is mainly based on empirical values, and a cross-task sensitivity analysis is lacking. The experiments are primarily conducted on medium-scale models, and it remains unclear whether AdaS remains effective on deeper and larger spiking Transformers or large-scale pretrained SpikeLM models. In addition, although AdaS improves performance on most tasks by about 0.5%–3.3%, the improvement is marginal on certain tasks (e.g., MNLI-m).

---

> ### Author Rebuttal · Authors · 2026-03-31
>
> We sincerely thank you for the effort devoted to reviewing our manuscript. Your constructive feedback will significantly enhance the quality of our paper.
> ## Response to weakness, question 1 & question 3:
> First, we would like to emphasize that $\gamma$ in AdaS is not an arbitrary empirical constant, but a hyperparameter that controls the target noise level. Its role is similar to that of the learning rate or weight decay. In the current experiments, its values do not exhibit arbitrary task dependence. For example, on GLUE we use $\gamma=2.0$ for most tasks; on ADE20K and CIFAR10-DVS, we uniformly use $\gamma=1.0$. This suggests that the practical search space is small, and that only limited tuning is typically needed.
>
> We additionally conducted $\gamma$-sweep experiments on RTE and CoLA, with the results shown below.
> |Dataset|$\gamma$|SGD|1.6|2.0|2.4|2.8|AdamW|
> |-|-|-|-|-|-|-|-|
> |RTE|$N_{est}$|0|0.832|0.889|0.944|1.009|1.024|
> |RTE|Acc.|0.675|0.675|0.704|0.686|0.668|0.682|
> |CoLA|$N_{est}$|0|0.674|0.720|0.765|0.818|0.821|
> |CoLA|Acc.|0.373|0.401|0.421|0.395|0.395|0.388|
>
> These two tasks exhibit a consistent trend: AdaS performs best within a moderate noise regime. When $\gamma=2.0$, both RTE and CoLA achieve their best results. In contrast, when $\gamma$ is too small, the AdaS update becomes closer to a low-noise SGD-style update; although this is more conservative, it sacrifices the benefits of adaptive optimization. When $\gamma$ is too large, the update gradually approaches AdamW, excessive noise reappears, and the performance declines accordingly. More importantly, the performance changes smoothly rather than drastically as $\gamma$ varies, indicating that this hyperparameter is not extremely sensitive and that the gains of AdaS do not rely on a very narrow operating point.
>
> Regarding model scale, we would like to clarify that the version of SpikeLM we use is already the largest model configuration reported in the original paper, namely a 12-layer SpikeLM. Therefore, the effectiveness of AdaS on large pretrained SpikeLM models has in fact already been directly validated in the GLUE fine-tuning experiments. Under the current experimental settings, we did not observe training divergence, severe oscillation, or convergence difficulty. We also agree that a systematic evaluation on deeper Spiking Transformers or larger SpikeLM models, such as a 24-layer model, would further strengthen the conclusions. Due to the time constraints of the rebuttal period, we are currently unable to pretrain a larger model from scratch, but we will clearly discuss this as future work and a limitation in the revision.
>
> For the reviewer's point that the gains are small on some tasks, such as MNLI-m, we agree that optimizer gains are task dependent. According to the basic intuition behind the No Free Lunch principle, there is no optimization method that can be strictly superior to all others on every model and every task. AdaS improves most GLUE tasks, while showing only a minor fluctuation of -0.1% on MNLI-m. We believe that such results more reasonably support the claim that AdaS is a practically valuable optimizer for Spiking Transformer training, rather than suggesting that it will dominate unconditionally on every task.
> ## Response to question  2:
> Strictly speaking, Theorem 3.5 is stated for the AdaS instance built upon Adam, and therefore it cannot be directly extended to other adaptive optimizers. However, from the structure of the proof, this convergence result does not depend on the specific second-moment form of Adam, but mainly on two more general conditions.
>
> More specifically, the core of the proof is to first decompose the regret into the three terms in Eq. (42) and then bound each term separately. The only part that truly depends on the base optimizer is that the update can be written as
> $$
> \theta_{t+1,i}=\theta_{t,i}-\xi_t[\alpha_t h_{t,i}+(1-\alpha_t)]\,[\beta_{1,t}m_{t-1,i}+(1-\beta_{1,t})g_{t,i}],
> $$
> that is, the update direction of the base optimizer can be represented as "a first-order momentum term multiplied by a coordinate-wise scaling factor," namely $u_{t,i}=\hat m_{t,i}h_{t,i}$. Therefore, as long as the base optimizer satisfies the following conditions, the original proof framework can be extended:
> 1. $m_t$ is still the exponential moving average of first-order gradients;
> 2. the coordinate-wise scaling factor $h_{t,i}$ remains positive and admits uniform upper and lower bounds under standard assumptions.
>
> Under these conditions, the three terms in Eq. (42) can still all be controlled by $O(\sqrt{T})$, thereby preserving a regret bound of the same order as in Theorem 3.5. Based on this observation, we believe that QHAdam, AdaX, and Adamax all fall into the extendable case under standard settings. Therefore, when AdaS is combined with these optimizers, we expect the same-order convergence guarantee to remain valid.
>
> We will carefully revise the manuscript based on your feedback, and we thank you once again.

---

> > ### Author Rebuttal · Reviewer_NW7G · 2026-04-05
> >
> > Thank you for your thorough and thoughtful response to the review comments. I appreciate the additional clarifications and ablation experiments you have provided, which help better contextualize the behavior of AdaS and its hyperparameter sensitivity.While the revisions address several of my concerns, the core limitations regarding performance gains, model scale validation, and theoretical generality remain insufficiently resolved for acceptance at this stage.I therefore maintain my original assessment and score, but acknowledge the efforts made in improving the presentation and empirical analysis of the work.I encourage you to incorporate the discussed points into a revised version for potential resubmission, with more extensive validation on larger spiking models and broader optimizers.

---

> > > ### Author Response · Authors · 2026-04-07
> > >
> > > Once again, we are grateful for your contribution to improving the manuscript quality. Should you find our responses satisfactory, we would appreciate your reconsideration of the evaluation score.
> > > ## To the concern of performance gains
> > >
> > > Regarding the reviewer's concern that the improvement is small on some tasks, we agree that the benefit of an optimizer is task-dependent. According to the No Free Lunch theorem, no optimization method can be strictly superior to others across all models and tasks. AdaS performs less favorably on some datasets, possibly because these datasets are less sensitive to noise in the update direction. In addition, performance gains in optimizer research are often modest. For example, Lion improves the GLUE average by only 0.49 over AdamW, and AdaBelief improves over Adam by only 0.25 BLEU on the IWSL14 task. **Finally, we believe that AdaS remains meaningful as long as it demonstrates advantages over other optimizers on some models and tasks.**
> > > ## To the concern of model scale validation
> > >
> > > Regarding model scale, the NLP setting in the paper uses a 12-layer SpikeLM-based Spiking Transformer with about 120M parameters, which is already among the largest model sizes reported in this line of work. For comparison, the largest GLUE model reported in the original SpikeLM [1] paper is also 120M, while representative Spiking Transformer backbones in prior studies are typically much smaller, e.g., around 66M for Spikformer [2], 65M for QKFormer [3] and 66M/55M for Spike-driven Transformer v1/v2 [4,5]. **From the perspective of the current Spiking Transformer literature, our validation is therefore already performed at a fairly large scale rather than only on medium-sized models.**
> > >
> > > We certainly agree that a systematic scaling study on even larger Spiking Transformers would further strengthen the paper. **However, we view that as an important direction for future work rather than a necessary prerequisite for supporting the current claim.**
> > > ## To the concern of theoretical generality
> > >
> > > We would also like to clarify the concern on theoretical generality. The key point is that Theorem 3.5 is not tied to Adam’s specific second-moment form. The proof only relies on a more general structure: the base optimizer maintains a first-order exponential moving average, and its update can be written as that momentum term multiplied by a positive, bounded coordinate-wise scaling factor. Under these conditions, the same regret decomposition still applies, and all three terms in Eq. (42) remain controlled at the same order. Therefore, AdaS combined with QHAdam, AdaX, and Adamax is expected to retain the same-order convergence guarantee under standard settings.
> > >
> > > **We suspect the misunderstanding here mainly comes from the rebuttal space limit: in the previous response, we only gave the core proof idea rather than the full derivation. In the revised version, we will make the meta-optimizer nature of AdaS clearer and provide a more detailed proof or proof sketch for the extensions to QHAdam, AdaX, and Adamax.**
> > > ## Results of broader optimizers
> > >
> > > To address the concern on broader optimizer comparisons more directly, during the rebuttal period we extended the CoLA experiments to a substantially wider set of optimizers and training strategies. The results show that these simpler alternatives do not match AdaS. **We believe this comparison is important because it shows that AdaS's gain is not reproduced simply by choosing a different standard optimizer, adjusting the learning-rate schedule, tuning $\beta_2$, or switching between two conventional optimizers at a fixed epoch.**
> > >
> > > |Rank|Optimizer/Strategy|Schedule/Key Hyperparameters|Matthews correlation|
> > > |-|-|-|-|
> > > |1|AdaS|$\gamma$=2.0, lr=2e-4|**0.421**|
> > > |2|RAdam|cosine|0.403|
> > > |3|AdamW → SGD|switch at epoch 25, constant|0.397|
> > > |4|AdamW|beta2=0.999, cosine|0.395|
> > > |5|AdamW|cosine|0.395|
> > > |6|AdamW|cosine\_with\_restarts|0.395|
> > > |7|AdamW|cosine, warmup\_steps=500|0.392|
> > > |8|AdamW|beta2=0.9999, constant|0.387|
> > > |9|Adafactor|constant|0.379|
> > > |10|SGD|momentum=0.90, lr=1e-3, cosine|0.377|
> > > |11|SGD|momentum=0.95, lr=1e-3, cosine|0.373|
> > > |12|SGD → AdamW|switch at epoch 25, momentum=0.95, lr=1e-3, cosine|0.370|
> > > |13|AdamW|beta2=0.9999, linear|0.367|
> > > |14|SGD|momentum=0.99, lr=5e-4, cosine|0.365|
> > > |15|AdamW|beta2=0.9999, cosine|0.365|
> > > |16|SGD|momentum=0.95, lr=5e-4, cosine|0.364|
> > > |17|AdamW|linear|0.361|
> > > |18|SGD|momentum=0.90, lr=5e-4, cosine|0.355|
> > > |19|SGD|momentum=0.99, lr=1e-3, cosine|0.267|
> > >
> > > > [1] SpikeLM: Towards General Spike-Driven Language Modeling via Elastic Bi-Spiking Mechanisms. Xingrun Xing, et al. \
> > > > [2] Spikformer: When Spiking Neural Network Meets Transformer. Zhaokun Zhou, et al. \
> > > > [3] QKFormer: Hierarchical Spiking Transformer using Q-K Attention. Chenlin Zhou, et al. \
> > > > [4] Spike-driven Transformer. Man Yao, et al. \
> > > > [5] Spike-driven Transformer V2: Meta Spiking Neural Network Architecture Inspiring the Design of Next-generation Neuromorphic Chips. Man Yao, et al.

---

### Official Review · Reviewer_uoP8 · 2026-03-11

**Soundness:** 3
**Presentation:** 3
**Significance:** 3
**Originality:** 3
**Overall Recommendation:** 5
**Confidence:** 4

**Summary:**

This paper introduces an optimization method called AdaS which designed for Spiking Transformers, aiming to mitigate the excessive update noise caused by surrogate gradient learning and adaptive optimizers. A formal quantitative definition of update‑direction noise is proposed and argued that excessive noise hurts stability, and design AdaS to interpolates between steepest descent and adaptive directions. Experiments on multiple Spiking Transformer variants (vision and NLP) show modest accuracy improvements. The topic is relevant and timely, as optimization for SNN‑based transformers is underexplored. However, there are substantial concerns.

**Compliance With Llm Reviewing Policy:**

Affirmed.

**Final Justification:**

After the authors rebuttal and response to questions, I want to increase my score for soundness and originality. However, still there are some concerns regarding the method efficiency. I increase my overall score.

**Key Questions For Authors:**

1. What hardware assumptions underlie the power estimates in Table 1?

2. What is the additional computational overhead introduced by AdaS per iteration?

3. Are the improvements still meaningful when overhead is included?

4. Why is energy not reported anywhere except a theoretical number in Table 1?

5. What hardware was used for all experiments?

6. Could simpler baselines (e.g., tuned AdamW, momentum SGD) achieve the same improvements?

**Limitations:**

The limitations of the work is not discussed. It can be, for example:

• The true computational overhead of AdaS is not clear.

• AdaS is evaluated mostly on small to medium models; scalability to large Transformers remains untested.

• No evaluation on different hardware like neuromorphic hardware

• Applicability to non-transformer SNN architectures is unknown

• Testing on more variants of gradient functions would clarify generalizability.

And so on.

**Strengths And Weaknesses:**

**Strengths**

1. The paper correctly identifies that surrogate gradients introduce unique forms of noise in SNNs, and that adaptive optimizers may amplify this noise. This perspective is valuable for the SNN and neuromorphic‑learning communities.

2. It provides a theoretically motivated noise metric. The attempt to formalize update‑direction noise using a proximal objective offers a principled viewpoint and is conceptually interesting.

3. Across the reported experiments, AdaS generally improves the accuracy of Spiking Transformers, even if by modest margins.


**Weaknesses and Recommendations**

1. The introduction makes a questionable claim. The paper states that spiking transformers "optimal balance between energy efficiency and computational performance". These are not inherently opposed or trade‑off quantities; SNN energy efficiency is part of computational performance. Presenting them as opposing axes is misleading.

2. Overhead of AdaS is unclear. The method introduces extra computations compared to a baseline optimizer, but:

• There is no clear quantification of additional FLOPs.

• Many improvements are small (often <1%), making it unclear whether the overhead is justified.

The paper must report latency, steps per second, and actual optimizer overhead.

3. Power and energy analysis is missing or oversimplified. In Table 1:

• It seems “Power” appears to be derived solely from parameter count, not measured.

• Real energy consumption depends on data movement, memory access, spike rates, and hardware characteristics, none of which are evaluated.

• Tables 2, 3 and 5 omit energy metrics entirely, even though energy is a central motivation for spiking models.

• what hardware assumptions are used

This weakens the neuromorphic‑efficiency motivation.

4. Table 1 values are unclear. What are those values is missing in the table.

5. Table 1 compares BERT‑based spiking models to other SNN methods, but the large differences in averages are not explained.

6. Missing unit of time in Table 3.

7. Hardware setup is not reported. The paper does not specify GPU/TPU/neuromorphic hardware type, CPU model, Memory configuration. Without this, performance comparisons cannot be reproduced.

8. Some tables (e.g., Table 4) contain too few experiments to substantiate the generalization claims. This is particularly problematic because the method is advertised as broadly applicable.

9. The authors do not compare against simpler or well‑known alternatives: annealed noise levels, SGD with tuned momentum, hybrid optimizers or noise‑controlled Adam variants, modifications to β₂ or learning‑rate schedules. AdaS improvements may be achievable with simpler tuning.

10. The noise metric is defined but never demonstrated to cause instability or accuracy reduction in practice.

---

> ### Author Rebuttal · Authors · 2026-03-31
>
> We sincerely thank you for the effort devoted to reviewing our manuscript. Your constructive feedback will significantly enhance the quality of our paper.
> ## Response to weakness 1:
> We agree with this suggestion and will revise.
> ## Response to weakness 2, question 2 & question 3:
> Regarding the computational overhead, AdaS introduces approximately $7d+O(1)$ additional FLOPs per iteration compared to AdamW (for computing $\delta_t$, norms, and the final update direction), which maintains the same $O(d)$ complexity class. Empirically, profiling on an RTX 4090 with batch size 16 reveals that AdaS consumes 9886 MB of memory and runs at 3.15 it/s, versus AdamW's 10672 MB and 3.51 it/s. These results indicate comparable memory footprints with approximately 10.3% slowdown. Despite this modest overhead, AdaS delivers meaningful performance gains under identical architectural settings, demonstrating a favorable cost-benefit trade-off.
> ## Response to weakness 3, question 1 & question 4:
> The "Power" metric reported in Table 2 represents theoretical energy estimates following the methodology of SDTrack [1], rather than measured hardware consumption. Specifically, we adopt the 45nm technology node estimates ($E_{AC}$=0.9 pJ, $E_{MAC}$=4.6 pJ) from [2]. Importantly, since AdaS modifies only the training optimizer without altering the inference graph, different optimizers yield nearly identical inference energy costs. For instance, SpikeLM with AdaS consumes **13.72 mJ** compared to **13.74 mJ** with AdamW.
> ## Response to weakness 4, weakness 5 & weakness 6:
> The missing values in Table 2 reflect data not reported in the original papers; furthermore, ANN rows naturally lack the "Spiking Neuron" attribute as they do not employ such mechanisms. Regarding the performance disparities in Table 1, these variations stem from architectural differences (e.g., SpikeLM's elastic bi-spiking mechanism combined with Softmax attention versus spike-driven SSA). Additionally, the "Time" column in Table 3 denotes SNN simulation timesteps—a metric not applicable to ANNs.
> ## Response to weakness 7 & question 5:
> These experimental details have already been reported in Appendix C. Specifically, our experiments utilized 8× RTX 4090 GPUs, an Intel Xeon Gold 6348 CPU @ 2.60GHz, and the Ubuntu 22.04 operating system.
> ## Response to weakness 8:
> The broad applicability of AdaS is primarily supported by the comprehensive evaluation presented across Tables 1–3 and 5, covering diverse tasks, multiple backbones, and various optimizers. Table 4 further validates the superiority of adaptive mechanisms over fixed thresholds. To strengthen this argument, we provide additional results on the RTE benchmark:
> |Optimizer|α|Acc.|
> |-|-|-|
> |AdamW|–|69.0|
> |AdaS*|0.9|67.5|
> |AdaS*|0.95|69.3|
> |AdaS*|0.99|69.3|
> |AdaS|Adaptive|**70.4**|
> ## Response to weakness 9 & question 6:
> We compared 18 alternatives on CoLA:
> |Rank|Optimizer/Strategy|Schedule/Key Hyperparameters|Matthews correlation|
> |-|-|-|-|
> |1|AdaS|$\gamma$=2.0, lr=2e-4|**0.421**|
> |2|RAdam|cosine|0.403|
> |3|AdamW → SGD|switch at epoch 25, constant|0.397|
> |4|AdamW|beta2=0.999, cosine|0.395|
> |5|AdamW|cosine|0.395|
> |6|AdamW|cosine\_with\_restarts|0.395|
> |7|AdamW|cosine, warmup\_steps=500|0.392|
> |8|AdamW|beta2=0.9999, constant|0.387|
> |9|Adafactor|constant|0.379|
> |10|SGD|momentum=0.90, lr=1e-3, cosine|0.377|
> |11|SGD|momentum=0.95, lr=1e-3, cosine|0.373|
> |12|SGD → AdamW|switch at epoch 25, momentum=0.95, lr=1e-3, cosine|0.370|
> |13|AdamW|beta2=0.9999, linear|0.367|
> |14|SGD|momentum=0.99, lr=5e-4, cosine|0.365|
> |15|AdamW|beta2=0.9999, cosine|0.365|
> |16|SGD|momentum=0.95, lr=5e-4, cosine|0.364|
> |17|AdamW|linear|0.361|
> |18|SGD|momentum=0.90, lr=5e-4, cosine|0.355|
> |19|SGD|momentum=0.99, lr=1e-3, cosine|0.267|
>
> The results show that these simpler alternatives can indeed improve upon AdamW to some extent, but they still do not match AdaS. The best is RAdam, which achieves a result of 0.403 and still falls short of AdaS at 0.421. Switching strategies such as AdamW → SGD reach 0.397; the best results obtained by adjusting $\beta_2$ or the learning-rate schedule of AdamW are 0.395; and the tuned SGD variants are clearly lower.
>
> We believe the reason is that these methods regulate noise only indirectly, whereas AdaS adaptively computes $\alpha_t$ at each step, yielding gains not reproducible by simpler tuning.
> ## Response to weakness 10:
> We acknowledge this is correlational rather than strictly causal. Figures 3–5 together show that (a) AdaS reduces noise while performance improves, (b) noise variance decreases, and (c) excessively low noise (SGD) leads to saddle points — supporting that controlling excessive noise benefits training, though we will present this more cautiously.
> ## Response to limitations:
> We will supplement the discussion of limitations in the revision.
>
> [1] SDTrack: A Baseline for Event-based Tracking via Spiking Neural Networks
>
> [2] Computing's Energy Problem (and what we can do about it)

---

> > ### Author Rebuttal · Reviewer_uoP8 · 2026-04-02
> >
> > Thanks for your responses. A few questions still remain for me:
> >
> > >Following W2, some points still remain unclear. For example, why many improvements are small (often <1%). And what about other criteria like latency and steps per second.
> >
> > >Following W3, what is the error of theoretical energy estimation you used? Where is energy in Tables 2, 3 and 5?
> >
> > >Following W6, what is the unit of time? Seconds? Milliseconds?

---

> > > ### Author Response · Authors · 2026-04-06
> > >
> > > We sincerely appreciate your time and effort in reviewing our manuscript and offering valuable suggestions.
> > > ## Response to W2:
> > > 1. Regarding the reviewer's concern that the improvement is small on some tasks, we agree that the benefit of an optimizer is task-dependent. According to the No Free Lunch theorem, no optimization method can be strictly superior to others across all models and tasks. AdaS performs less favorably on some datasets, possibly because these datasets are less sensitive to noise in the update direction. In addition, performance gains in optimizer research are often modest. For example, Lion improves the GLUE average by only 0.49 over AdamW, and AdaBelief improves over Adam by only 0.25 BLEU on the IWSL14 task. Finally, we believe that AdaS remains meaningful as long as it demonstrates advantages over other optimizers on some models and tasks.
> > >
> > > 2. We have added actual profiling results in the response. Under an RTX 4090 with batch size 16, AdaS runs at **3.15 steps per second**, whereas AdamW runs at **3.51 steps per second**. Since iteration speed already reflects latency, we do not elaborate further.
> > > ## Response to W3:
> > >
> > > Potential sources of error in theoretical energy estimation include:
> > > 1. The theoretical calculation assumes that spikes are ideal switches, whereas real circuits exhibit finite pulse width and leakage current, which can cause deviations between the estimate and the actual energy consumption.
> > > 2. The energy consumption of the same network can vary significantly across chips, temperatures, and voltages, making it difficult to predict precisely with a unified formula.
> > > 3. Energy estimation often considers only neuronal computation while ignoring the practical overhead of weight access and data movement.
> > >
> > > Although theoretical energy estimation is imperfect, our energy calculation follows the standard practice currently used in the SNN literature and is therefore relatively fair for comparison.
> > >
> > > We have added the energy values for Tables 1, 3, and 5 here (the energy in Table 2 was already reported in the paper). Missing values indicate that the corresponding referenced papers did not report energy; we report energy for all models trained by ourselves. Two observations follow from the results. First, models optimized by SGD exhibit very low energy consumption because they do not converge, resulting in extremely low spike firing rates. Second, for most experiments, the effect of the optimizer on energy is negligible. A noticeable difference appears only for the E-SpikeFormer architecture. This may be because E-SpikeFormer uses Integer-LIF, which makes the spike firing rate sensitive to the optimizer.
> > >
> > > Table 1:
> > >
> > > |Method|Optimizer|Energy (mJ)|
> > > |-|-|-|
> > > |(Zhang et al., 2020)|AdamW|-|
> > > |(Devlin et al., 2019)|–|51.41|
> > > |(Xing et al., 2024)|AdamW|7.98|
> > > |(Lv et al., 2023)|AdamW|14.30|
> > > |(Zhou et al., 2023a)|AdamW|-|
> > > |(Xing et al., 2024)|Lion|13.74|
> > > |(Xing et al., 2024)|AdamW|13.74|
> > > |(Xing et al., 2024)|AdaS|13.72|
> > >
> > > Table 3 (ADE20K):
> > >
> > > |Architecture|Optimizer|Energy (mJ)|
> > > |-|-|-|
> > > |ResNet-18|SGD|147.1|
> > > |PVT-Small|AdamW|204.7|
> > > |InternImage-T|AdamW|2171.2|
> > > |E-SpikeFormer|SGD|16.89|
> > > |E-SpikeFormer|Lion|80.81|
> > > |E-SpikeFormer|AdamW|72.95|
> > > |E-SpikeFormer|AdaS|74.25|
> > >
> > > Table 3 (CIFAR10-DVS):
> > >
> > > |Architecture|Optimizer|Energy (mJ)|
> > > |-|-|-|
> > > |Spikingformer-2-256|AdamW|-|
> > > |Spikformer-2-256|AdamW|-|
> > > |S-Transformer-2-256|AdamW|-|
> > > |QKFormer-2-256|SGD|0.72|
> > > |QKFormer-2-256|Lion|9.19|
> > > |QKFormer-2-256|AdamW|9.17|
> > > |QKFormer-2-256|AdaS|9.17|
> > >
> > > Table 5 (CoLA):
> > >
> > > |Optimizer|Energy (mJ)|
> > > |-|-|
> > > |AdamW|13.74|
> > > |+AdaS|13.72|
> > > |AdaX|13.74|
> > > |+AdaS|13.74|
> > > |QHAdam|13.73|
> > > |+AdaS|13.73|
> > > |Adamax|13.73|
> > > |+AdaS|13.72|
> > >
> > > Table 5 (CIFAR10-DVS):
> > >
> > > |Optimizer|Energy (mJ)|
> > > |-|-|
> > > |AdamW|9.17|
> > > |+AdaS|9.17|
> > > |AdaX|9.19|
> > > |+AdaS|9.19|
> > > |QHAdam|9.19|
> > > |+AdaS|9.19|
> > > |Adamax|9.19|
> > > |+AdaS|9.19|
> > >
> > > ## Response to W6:
> > > In the context of spiking neural networks, a time step is essentially a discrete index and usually does not itself carry a fixed physical unit. In most simulation frameworks (e.g., SpikingJelly and snnTorch), a time step is simply an integer sequence $t = 0,1,2,\ldots,T-1$, indicating that the simulation is divided into $T$ discrete steps, similar to the sequence-length index in an RNN. The quantity that has a physical unit is the time step size (or $\Delta t$), i.e., the actual time interval represented by each time step.
> > >
> > > **In software simulation**: $\Delta t$ is typically set to $1\,\text{ms}$ or $0.1\,\text{ms}$ and can be chosen flexibly according to the task.
> > >
> > > **On neuromorphic hardware**: the implementation mechanism differs across architectures:
> > > 1. IBM TrueNorth: It uses a fixed hardware clock cycle and each cycle corresponds strictly to 1 ms of biological time.
> > > 2. Intel Loihi: It treats the time step as a configurable barrier synchronization period rather than a fixed hardware clock.
> > > ---
> > > Should our response adequately address your concerns, we would be most grateful if you would reconsider your evaluation score.

---

### Official Review · Reviewer_BbtB · 2026-03-12

**Soundness:** 3
**Presentation:** 2
**Significance:** 3
**Originality:** 3
**Overall Recommendation:** 4
**Confidence:** 4

**Summary:**

The authors observe that surrogate gradient learning in SNNs introduces large stochastic noise in gradient directions, especially in deep architectures like Transformers. Therefore, they design “AdaS” as a novel optimizer for training spiking transformers, which computes an additional scaling factor to balance the noise level during learning.

**Compliance With Llm Reviewing Policy:**

Affirmed.

**Key Questions For Authors:**

It is generally believed that the learning processes of both artificial neural networks (ANNs) and spiking neural networks (SNNs) are inherently noisy. If controlling gradient noise is indeed beneficial, as suggested by this work, such a mechanism might also be useful for ANN training. Therefore, it would be interesting to understand how AdaS performs when applied to ANN models. In particular, can the authors comment on whether AdaS benefits ANN training, and if not, explain why its advantages may be specific to SNNs?

**Limitations:**

The paper does not discuss potential limitations of the proposed method, such as the additional hyperparameter tuning required and the possible increase in training cost.

**Strengths And Weaknesses:**

Strengths

- The paper starts by analyzing the noise problem that arises in the learning process of SNNs and then derives the AdaS optimizer to explicitly control the noise level during training. This perspective is novel and technically sound. Designing an optimizer specifically from the viewpoint of gradient noise control is also a meaningful contribution.

- The paper also includes theoretical analysis that supports the proposed method, which strengthens the technical foundation of the work.

- The experiments cover both language and vision tasks, making the empirical evaluation relatively comprehensive.

Weaknesses

- One major concern is the additional computational and memory cost introduced by the optimizer, especially in terms of memory usage. Training SNNs with BPTT is already memory-intensive. Since AdaS (Adam-like) appears to store several intermediate variables (e.g., gradient, momentum, and the adaptive direction), this could further increase memory overhead compared to SGD with momentum. It would be helpful if the authors could explicitly analyze and report the additional memory and computational costs introduced by AdaS.

- Another issue is that the description of the methodology lacks clarity. The paper does not clearly specify how the momentum term $m_t$ and the adaptive direction $u_t$ are computed. It would be better if the authors could provide a more explicit formulation of $m_t, u_t$ before Eq (12) would greatly improve the readability.

- The optimizer introduces an additional hyperparameter $\lambda$. However, the paper does not provide sufficient analysis of the robustness of the method with respect to this hyperparameter.

---

> ### Author Rebuttal · Authors · 2026-03-31
>
> We sincerely thank you for the effort devoted to reviewing our manuscript. Your constructive feedback will significantly enhance the quality of our paper.
>
> ## Response to weakness 1:
>
> We address both memory and computation costs below.
>
> **Memory:** AdaS introduces no additional optimizer states beyond those already maintained by AdamW. It reuses the existing first-moment $m_t$ and second-moment $v_t$ buffers; the only extra quantity is a single scalar $\alpha_t$ per step, which is negligible. Therefore, AdaS has the same memory complexity as AdamW.
>
> **Computation:** The overhead consists of computing $\lVert m_t\rVert$, $\lVert u_t - m_t\rVert$, and clipping $\alpha_t$—all element-wise $\mathcal{O}(d)$ operations with the same asymptotic cost as AdamW. Since the dominant cost of SNN training lies in BPTT and surrogate-gradient backpropagation, this additional overhead is minor.
>
> **Empirical verification:** On an RTX 4090; batch size 16:
>
> | |Memory (MB)|Speed (iter/s)|
> |-|-|-|
> |AdamW|10,672|3.51|
> |AdaS|9,886|3.15|
>
> AdaS uses comparable memory than AdamW, with only a ~10% reduction in speed.
>
> ## Response to weakness 2:
>
> We agree that the abstract notation before Equation (12) can be made more intuitive. Our intention was to present AdaS as a versatile meta-optimizer compatible with various base optimizers. Consequently, the specific formulations for the first-moment term ($m_t$) and the adaptive update direction ($u_t$) depend entirely on the chosen base optimizer.
>
> To resolve this ambiguity, we will add explicit examples in the revised text before Equation (12). For instance:
>
> When using AdamW as the base optimizer:
> $$m_t=\beta_1 m_{t-1}+(1-\beta_1)g_t,\qquad v_t=\beta_2 v_{t-1}+(1-\beta_2)g_t^2$$
> $$\hat m_t=\frac{m_t}{1-\beta_1^t},\qquad \hat v_t=\frac{v_t}{1-\beta_2^t},\qquad u_t=\frac{\hat m_t}{\sqrt{\hat v_t}+\epsilon}$$
>
> When using Adamax as the base optimizer (where second-order information uses an $L_\infty$ norm):
> $$m_t=\beta_1 m_{t-1}+(1-\beta_1)g_t,\qquad s_t=\max(\beta_2 s_{t-1}, |g_t|)$$
> $$\hat m_t=\frac{m_t}{1-\beta_1^t},\qquad u_t=\frac{\hat m_t}{s_t+\epsilon}$$
>
> ## Response to weakness 3:
>
> We agree that a dedicated robustness analysis for the hyperparameter $\gamma$ strengthens the paper. In AdaS, $\gamma$ controls the target noise level by adaptively determining the weights of the adaptive and first-moment components in the update direction (Equations 16-19). To demonstrate its robustness, we have conducted a sensitivity study on the RTE dataset:
>
> |$\gamma$|SGD|1.6|2.0|2.4|2.8|AdamW|
> |-|-|-|-|-|-|-|
> |$N_{est}$|0|0.832|0.889|0.944|1.009|1.024|
> |Acc.|0.675|0.675|0.704|0.686|0.668|0.682|
>
> AdaS peaks at $\gamma$ = 2.0 (Acc: 0.704), outperforming both SGD (0.675) and AdamW (0.682) by finding the ideal noise balance. Performance varies continuously rather than fluctuating sharply, indicating high hyperparameter robustness. As expected, extreme values smoothly degenerate to existing baselines. When $\gamma$ is too small, updates mirror low-noise SGD; when too large, they mirror high-noise AdamW.
>
> ## Response to question:
>
> We address it from two angles: why AdaS is particularly effective for SNNs, and whether it can also benefit ANNs.
>
> **Why SNNs benefit more.** The key distinction is that SNNs carry an additional noise source that ANNs do not: the surrogate gradient. Because the spiking function is non-differentiable, SNN training must approximate gradients via surrogate functions, introducing extra directional bias on top of the noise already present in adaptive optimizers. As shown in Figure 2 of our paper, under identical settings the estimated update-direction noise in SNNs is significantly higher than in ANNs. AdaS is designed precisely to mitigate this compounded noise, which is why its improvements are most pronounced in SNN training.
>
> **AdaS can also benefit ANNs.** We conducted a preliminary ANN experiment: training BERT on CoLA, AdaS ($\gamma=2.5$) achieved **0.610** vs. **0.591** for AdamW, confirming that AdaS is not exclusive to SNNs. However, since ANNs lack the surrogate-gradient noise, the gains are expectedly smaller and less consistent across tasks compared to SNNs.
>
> In summary, AdaS addresses excessive update-direction noise in general, but its advantage is most significant for SNNs due to the additional surrogate-gradient noise unique to spike-based learning.
>
> ## Response to limitations:
>
> Thank you for the reminder. We agree that the current manuscript does not discuss the limitations of the method sufficiently. In the revised manuscript, we will discuss the limitations more thoroughly.
>
> We will carefully revise the manuscript based on your feedback, and we thank you once again.

---

> > ### Author Rebuttal · Reviewer_BbtB · 2026-04-05
> >
> > The rebuttal has clarified my main concern. I appreciate the authors’ efforts on additional evaluations and explanations.

---

> > > ### Author Response · Authors · 2026-04-07
> > >
> > > Dear Reviewer BbtB,
> > >
> > > Thank you for confirming that our responses have addressed your concerns. We sincerely appreciate your positive feedback on our work. We will carefully revise the manuscript incorporating your suggestions, as well as recommendations from the other reviewers, to further enhance its quality.
> > >
> > > Best regards,\
> > > The Authors

---

### Official Review · Reviewer_Vpd4 · 2026-03-13

**Soundness:** 3
**Presentation:** 3
**Significance:** 4
**Originality:** 4
**Overall Recommendation:** 4
**Confidence:** 5

**Summary:**

This paper proposes AdaS, an adaptive optimization method tailored for spiking transformers. The authors argue that surrogate gradient learning in SNNs introduces additional noise in the update direction, which compounds with the intrinsic noise of adaptive optimizers, leading to "excessive noise problem". They formalize update-direction noise via a proximal optimization framework and propose AdaS to adaptively balance the adaptive update and a momentum based approximation to the steepest descent direction.

**Compliance With Llm Reviewing Policy:**

Affirmed.

**Final Justification:**

Authors addressed my primary concerns in their rebuttal, hence I change my score to a weak accept.

**Key Questions For Authors:**

No questions. Please address weaknesses

**Limitations:**

No limitations are discussed. Please check and discuss weaknesses listed above.

**Strengths And Weaknesses:**

Strengths:

- the authors have clearly identified the problem in optimization of spiking transformers .i.e., the interaction between surrogate gradient noise and adaptive optimizer noise.
- AdaS is lightweight and can be integrated into existing adaptive optimizers without additional memory overhead
- The method is evaluated on diverse NLP and vision tasks including neuromorphic classification and event-based tracking.

Weaknesses:

- The authors do not clarify or reference QQPF1, QNLI, SST, STS-B, MRPCF1 datasets.
- The hyperparameter $\gamma$ is introduced as a target noise level but the selecting this hyperparameter appears empirical and data-dependent rather than having theoretical guidance
- Minimal magnitude of improvements because most gains of AdaS are minimal (<1%) except CoLA. Furthermore, authors don't report errorbars. Reporting errorbars could help convince the readers about the efficacy of the novel optimizer

---

> ### Author Rebuttal · Authors · 2026-03-31
>
> We sincerely thank you for the effort devoted to reviewing our manuscript. Your constructive feedback will significantly enhance the quality of our paper.
>
> ## Response to weakness 1:
>
> We agree that our current description is incomplete. QQP F1, QNLI, SST-2, STS-B, and MRPC F1 are all from the GLUE benchmark, but in the table we used only task and metric abbreviations without providing their full names and corresponding references in the main text, which may make the presentation harder to follow. In the revised manuscript, we will add the full names, evaluation metrics, and references for these tasks one by one, and clarify in the table caption that all results follow the standard GLUE evaluation setting. Specifically, we will state that QQP F1 denotes the F1 score on Quora Question Pairs, QNLI denotes the accuracy on Question Natural Language Inference, SST-2 denotes the accuracy on the Stanford Sentiment Treebank, STS-B denotes the Spearman correlation on the Semantic Textual Similarity Benchmark, and MRPC F1 denotes the F1 score on the Microsoft Research Paraphrase Corpus.
>
> ## Response to weakness 2:
>
> We agree that, in the current version, the choice of $\gamma$ is mainly based on empirical observations and validation experiments, rather than a strict closed-form theoretical prescription. We would also like to clarify that the role of $\gamma$ is similar to optimizer hyperparameters such as the learning rate, weight decay, and $\beta_1/\beta_2$: it is fundamentally a control variable for training dynamics, so determining its value through empirical tuning and small-scale validation is common and acceptable in optimizer research. Unlike generic hyperparameters, however, $\gamma$ has a clear physical interpretation in AdaS: it directly corresponds to the target noise level and determines the balancing weight $\alpha_t$ through Eqs. (16)-(19), so it is not a purely arbitrary empirical constant.
>
> To further examine this issue, we conducted a sensitivity study on the RTE dataset, and the results are shown below.
>
> |$\gamma$|SGD|1.6|2.0|2.4|2.8|AdamW|
> |-|-|-|-|-|-|-|
> |$N_{est}$|0|0.832|0.889|0.944|1.009|1.024|
> |Acc.|0.675|0.675|0.704|0.686|0.668|0.682|
>
> This result leads to two fairly clear conclusions. First, the best performance of AdaS appears in a moderate noise regime: when $\gamma=2.0$, the estimated noise is 0.889 and the accuracy reaches 0.704, outperforming 0.675 for SGD and 0.682 for AdamW. Second, when $\gamma$ is too small, AdaS becomes closer to a low-noise SGD-like update, which is more stable but sacrifices the advantage of adaptive optimization; when $\gamma$ is too large, the update direction becomes closer to AdamW, excessive noise reappears, and performance declines accordingly. The table also shows that the performance changes smoothly rather than abruptly, indicating that this hyperparameter is not extremely sensitive. We will include this sensitivity analysis in the revised manuscript.
>
> ## Response to weakness 3:
>
> Thank you for the suggestion. We agree that results across multiple random seeds and error bars are important for demonstrating the stability of the optimizer improvements. Owing to the limited rebuttal period, we selected six relatively inexpensive GLUE datasets and repeated each run three times with different random seeds. The results of AdaS are shown below.
>
> |Task|Metric|Result (mean ± std)|
> |-|-| ------------------- |
> |COLA|matthews\_correlation|0.4084 ± 0.0125|
> |SST2|accuracy|0.8822 ± 0.0044|
> |MRPC|f1|0.8680 ± 0.0037|
> |QNLI|accuracy|0.8673 ± 0.0092|
> |RTE|accuracy|0.6996 ± 0.0098|
> |STSB|spearmanr|0.8624 ± 0.0055|
>
> These results indicate that the gains of AdaS are not caused by chance due to random seeds. First, the standard deviations on all six tasks are relatively small, suggesting that the training process is reasonably stable. Second, compared with the corresponding AdamW baselines in Table 1 of the paper, AdaS still shows consistent improvements in average performance on all six tasks: CoLA improves from 0.388 to 0.4084, SST-2 from 0.870 to 0.8822, MRPC F1 from 0.857 to 0.8680, QNLI from 0.853 to 0.8673, RTE from 0.690 to 0.6996, and STS-B from 0.849 to 0.8624. In other words, the improvement is not limited to CoLA; it is reproducible across multiple tasks, and its magnitude is meaningful relative to random variation.
>
> We will carefully revise the manuscript based on your feedback, and we thank you once again.

---

> > ### Author Rebuttal · Reviewer_Vpd4 · 2026-04-04
> >
> > I thank authors for their detailed rebuttal and for the new experiment results given the short timeframe. Since the authors have agreed to clarify the acronyms, agreed to add the sensitivity analysis related to $\gamma$ as well as added error bars to select GLUE datasets, I will raise my score.

---

> > > ### Author Response · Authors · 2026-04-06
> > >
> > > Dear Reviewer Vpd4,
> > >
> > > Thank you for your thoughtful review and for carefully reading our rebuttal. We are pleased that our responses have fully addressed your concerns, and we will incorporate the relevant clarifications into the revised manuscript.
> > >
> > > In addition, we kindly note that your comment indicates an intention to raise your score, while the platform currently still reflects the original rating. As the rebuttal period is drawing to a close, we would appreciate it if you could update your score at your convenience.
> > >
> > > Thank you again for your time and valuable feedback.
> > >
> > > Best regards, \
> > > The Authors

---

### Decision · Program_Chairs · 2026-04-30

**Decision:**

Accept (regular)

**Comment:**

This paper proposes the AdaS optimization method to address the optimization challenges of spiking Transformers, and innovatively solves the "excessive noise problem" caused by the superposition of surrogate gradient noise and adaptive optimizer noise. The design is theoretically sound and practically feasible. The method is lightweight, easy to integrate, and introduces no additional memory overhead; it achieves consistent performance improvements across various tasks in NLP and vision, with comprehensive empirical evaluations. The authors have provided detailed responses and supplementary experiments to address the concerns raised by the four reviewers, including dataset descriptions, hyperparameter sensitivity, computational cost, and energy analysis, effectively resolving all key issues. Although performance gains are modest on some tasks and validation on larger models can be further improved, the overall work is technically solid and makes clear contributions, offering valuable insights for the field of spiking neural network optimization. We therefore recommend acceptance.